# Offline Deep Reinforcement Learning for Visual Distractions via Domain Adversarial Training

**Jen-Yen Chang**                                          *chou@mi.t.u-tokyo.ac.jp*
*Graduate School of Information Science and Technology, The University of Tokyo*

**Thomas Westfectel**                                      *thomas@mi.t.u-tokyo.ac.jp*
*Graduate School of Information Science and Technology, The University of Tokyo*

**Takayuki Osa**                                           *osa@mi.t.u-tokyo.ac.jp*
*Graduate School of Information Science and Technology, The University of Tokyo*
*RIKEN Center for Advanced Intelligence Project (AIP)*

**Tatsuya Harada**                                         *harada@mi.t.u-tokyo.ac.jp*
*Graduate School of Information Science and Technology, The University of Tokyo*
*RIKEN Center for Advanced Intelligence Project (AIP)*

**Reviewed on OpenReview:** *https://openreview.net/forum?id=dce6ZGkJ1Z*

## Abstract

Recent advances in offline reinforcement learning (RL) have relied predominantly on learning from proprioceptive states. However, obtaining proprioceptive states for all objects may not always be feasible, particularly in offline settings. Therefore, RL agents must be capable of learning from raw sensor inputs such as images. However, recent studies have indicated that visual distractions can impair the performance of RL agents when observations in the evaluation environment differ significantly from those in the training environment. This issue is even more crucial in the visual offline RL paradigm, where the collected datasets can differ drastically from the testing environment. In this work, we investigated an adversarial-based algorithm to address the problem of visual distraction in offline RL settings. Our adversarial approach involves training agents to learn features that are more robust against visual distractions. Furthermore, we proposed a complementary dataset to add to the V-D4RL distraction dataset by extending it to more locomotion tasks. We empirically demonstrate that our method surpasses state-of-the-art baselines in tasks on both the V-D4RL and proposed dataset when evaluated on random visual distractions.

## 1 Introduction

Common model-free (Kumar et al., 2020; Fujimoto & Gu, 2021; An et al., 2021; Kostrikov et al., 2022) and model-based (Kidambi et al., 2020; Guo et al., 2022) offline reinforcement learning (RL) algorithms rely on learning from the proprioceptive states to address continuous control tasks. However, obtaining proprioceptive states is almost impractical in certain scenarios, such as large-scale environments, where defining the states of all objects is nearly unfeasible. Therefore, RL agents must learn from raw sensory inputs such as images. Training offline RL agents based on visual observations provides opportunities to make RL more widely applicable to real-world settings. Unlike offline RL agents trained from proprioceptive states, studies on offline RL agent training using visual observations (Lu et al., 2023) for continuous control tasks and well-designed benchmarks are scarce.

Frequently, using a visual encoder, we can estimate the latent proprioceptive features of an image-based observation and use these latent features to train the RL backbone algorithms (Hansen et al., 2021a; Yarats et al., 2022). Although this well-established approach is effective in scenarios with static backgrounds, it has

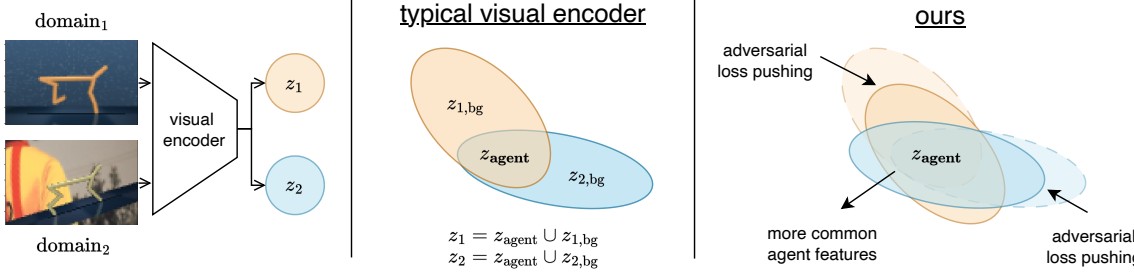

Figure 1: Our problem formulation. The agent is trained on two types of domains: a normal observation and a distracted observation. Intuitively, we can say that the latent features consist of agent features and distraction features. The visual encoder is incentivised by the adversarial training loss to extract latent features that are common to both domains by suppressing domain-specific visual distractions.

limitations when confronted with visual distractions. These distractions may include changes in background elements, viewpoints, or variations in the agent's colour scheme. Existing RL agents, whether offline or online, underperform in the presence of random visual distractions (Cobbe et al., 2020; Stone et al., 2021; Wang et al., 2021; Dupuis et al., 2022). By contrast, humans possess a remarkable ability to disregard visual distractions when observing actions from a visual input. This adeptness highlights our capacity to concentrate on domain-invariant features, focusing on aspects that remain consistent across diverse settings, such as an agent's movement patterns.

We hypothesised that existing vision-based RL algorithms are susceptible to random visual distractions because the visual encoder in these RL algorithms tends to overfit certain visual distractions during training. Consequently, these algorithms struggle to estimate robust latent features when exposed to unseen visual distractions during the evaluation. Our objective is to train the visual encoder in RL agents such that they can learn domain-invariant features within offline RL settings.

In this work, we investigated an adversarial learning approach to address the visual distraction issue in offline RL settings by extending a technique originally developed for domain adaptation (Ganin & Lempitsky, 2015). One important assumption is the accessibility to datasets from two types of domains, such as a dataset without distraction and another dataset with visual distractions.

We consider a framework where the agent is trained on two types of domains and subsequently evaluated on other unseen visual distractions. We introduce a domain discriminator that attempts to classify the latent features estimated by a visual encoder into two classes. The visual encoder is trained to minimise the discrepancy between the latent features of the two domains, whereas the domain discriminator aims to maximise this discrepancy. The problem formulation and schematics of the adversarial training method are presented in Figure 1. The visual encoder is incentivised by the adversarial training loss to extract latent features common to both domains by suppressing domain-specific visual distractions. Furthermore, we empirically demonstrated that incorporating DropBlock (Ghiasi et al., 2018) into a visual encoder is essential for improving the robustness of the proposed domain-adversarial training scheme.

To investigate the performance of offline RL algorithms in the presence of unseen distractions, we empirically evaluated existing offline RL algorithms on the V-D4RL distraction dataset. For a more comprehensive comparison, we collected offline datasets from additional locomotion tasks. We observed that these agents performed well when evaluated with previously seen distractions present in the training dataset, but exhibited a significant drop in performance when encountering unseen distractions during evaluation. Figure 2 illustrates this phenomenon. In summary, the contributions of this work are as follows:

1. We empirically demonstrated that commonly used offline RL methods are frequently susceptible to visual distractions. This highlights the need to develop techniques to enhance policy robustness against visual distractions.

2. We adopted an adversarial algorithm to visual-based offline RL settings and address the problem of visual distraction. The adversarial algorithm allows the visual encoder to learn domain-invariant features. Furthermore, we introduce DropBlock (Ghiasi et al., 2018) into our proposed offline RL algorithm for further robustness against unseen visual distractions.

3. We extended the existing V-D4RL distraction dataset, which contains only one visual distraction task, to four additional locomotion tasks for a comprehensive benchmarking of visual distractions between offline RL algorithms.

4. We empirically demonstrate that the adopted domain-adversarial learning method improves performance. Additionally, we compare our method with state-of-the-art offline RL algorithms and related online-based RL algorithms and empirically demonstrate that our method is more robust in visual distraction settings in both the V-D4RL dataset and our proposed novel dataset.

## 2 Related Work

### 2.1 Domain Invariance in RL

Recent advancements in RL, such as RAD (Laskin et al., 2020), DrQ (Yarats et al., 2021), and DrQv2 (Yarats et al., 2022), have demonstrated the effectiveness of data augmentation techniques in improving the RL performance, sample efficiency, and generalisation. Particularly in vision-based RL, data augmentation has emerged as a prominent strategy for reducing the reliance on specific training domains. These augmentation schemes enhance the robustness of the encoder representations and subsequently improve the policy, yielding superior results compared with previous methods (Yarats et al., 2022). For online visual-based RL, (Cobbe et al., 2019) investigated the effectiveness of common computer vision techniques for improving RL generalisation, and (Li et al., 2021) employed a gradient reversal layer in online RL settings.

Various visual-based generalisation methods include PAD (Hansen et al., 2021b) which employs self-supervised learning during evaluation; SVEA (Hansen et al., 2021a) which updates typical CNNs in encoders to transformers and uses stronger image augmentation; and ILA (Yoneda et al., 2022) which uses dynamic models to adapt during evaluation. In a broader context, Igl et al. (2019); Islam et al. (2023) proposed using the variational information bottleneck theory to enhance domain generalisation in RL. They introduced an additional policy gradient objective to simultaneously minimise the mutual information between different inputs, alongside actor-critic methods, thereby achieving better feature robustness. Our work focuses specifically on the robustness of the visual encoder, distinguishing it from previous methods. Domain invariance is also an important topic from the RL safety perspective, and (Haider et al., 2021) provided an overview and demonstrated how RL agents struggle to provide clear safety boundaries when deployed on safety-related tasks.

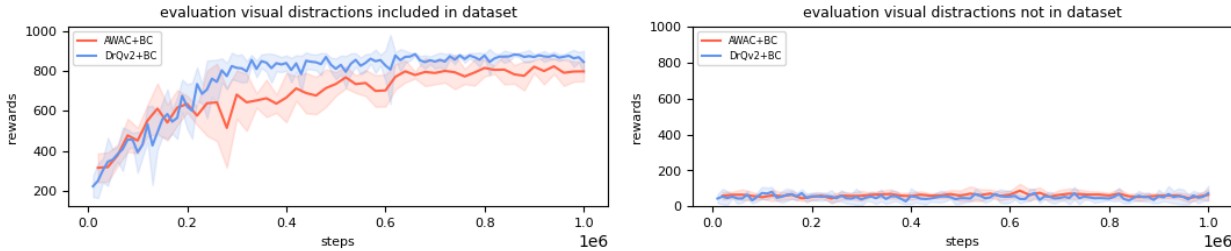

Figure 2: Initial result of baseline agents DrQv2+BC and AWAC+BC trained on the V-D4RL cheetah-run medium-expert easy-distraction dataset (Lu et al., 2023). When evaluating on the seen visual distractions that can be found in the training dataset, we can observe offline agents achieving expert-level performance (left). However, the agents perform very poorly when exposed to unseen random visual distractions (right).

## 2.2 Benchmarks for Offline Continuous Control

D4RL (Fu et al., 2020) is a prominent benchmark for continuous control of proprioceptive states in offline RL. The large variety of data distributions has allowed for the comprehensive benchmarking of state-of-the-art offline RL algorithms (Kidambi et al., 2020; Kostrikov et al., 2022; Kumar et al., 2020) and an understanding of the strengths and weaknesses of different tasks. Florence et al. (2022) discusses when offline RL algorithms would outperform behavioural cloning (BC) in learning from proprioceptive states. Conversely, vision-based datasets for discrete control were created for Atari by (Agarwal et al., 2020); however, they contained only 50M samples per environment. Hence, V-D4RL (Lu et al., 2023) aims to develop a comprehensive benchmark for visual-based continuous-control tasks. V-D4RL's 100 K benchmark represents a significantly more approachable challenge. Another issue with the current datasets is that they are all simulation-based. Zhou et al. (2022) proposed using data collected in real-world settings and demonstrated its suitability for robot learning in realistic environments. However, visual distraction data are lacking in the original V-D4RL dataset. Our novel datasets aimed to complement the V-D4RL dataset for visual distractions, which originally included only one task.

## 2.3 Dropout and its variants in RL

The application of dropout (Srivastava et al., 2014) to the RL domain is not new. For instance, Hiraoka et al. (2022) employed dropout in a critic network, demonstrating improved sample efficiency and achieved comparable performance with only two critic networks compared to the ten critic networks required in REDQ (Chen et al., 2021). Jaques et al. (2019) deployed dropout to obtain uncertainty estimates of the target Q values to alleviate the Q-learning overestimation bias. Unlike prior studies that focused on critic networks to enhance sample efficiency, this study targets the encoder as a source of improvement. To the best of our knowledge, this is one of the first studies to apply DropBlock (Ghiasi et al., 2018) to RL.

# 3 Preliminaries

## 3.1 Reinforcement Learning from Images

A standard RL problem for image-based control can be defined as an infinite-horizon Markov Decision Process MDP $= \langle \mathcal{S}, \mathcal{A}, p, \mathcal{R}, \gamma \rangle$, where $\mathcal{S}$ is the set of observations, $\mathcal{A}$ is the set of actions, $p$ is the transition probability function, $\mathcal{R}$ is the reward function, and $\gamma \in (0, 1)$ is the discount factor for future rewards. Generally, in image settings, image rendering of the system is not sufficient to fully describe the underlying state of the system. To this end, and per common practice (Mnih et al., 2013), we approximated the current observation of the system by stacking three consecutive prior images as an observation. We define the replay buffer $\mathcal{D}$ containing the observation, action, reward, and next observation at time step $t$ as $\mathcal{D} = (s_t, a_t, r_t, s_{t+1})$. The RL agent aims to maximise the discounted expected return $\mathbb{E}_\pi[\sum_{t=0}^{\infty} \gamma^t R(s_t, a_t)]$, which is the expected cumulative sum of rewards when following the policy in the MDP, where the importance of the horizon is determined by $\gamma$. Consequently, the goal is to determine a policy $\pi$ that maximises discounted expected returns.

## 3.2 Visual Actor-Critic Methods

We consider an architecture in which the input observation $s$ is first transformed using a data augmentation scheme. Subsequently, the encoder $f$ with parameters $\theta_f$ maps $s$ to a lower-dimensional latent feature vector $z = f(s; \theta_f)$. Q-networks have parameters $\theta_q$. For the replay buffer $\mathcal{D}$ and policy $\pi$, the policy objective becomes:

$$\pi = \underset{\pi}{\mathrm{argmax}} \mathbb{E}_{s_t \sim \mathcal{D}, a_t^\pi \sim \pi(\cdot | f(s_t; \theta_f))} \left[ Q(f(s_t; \theta_f), \ a_t^\pi; \ \theta_q) \right] \tag{1}$$

We employ clipped double Q-learning (Fujimoto et al., 2018) to reduce overestimation bias in the target value following previous works (Yarats et al., 2021; 2022; Lu et al., 2023). Target Q networks are Q networks with parameters $\overline{\theta_q}$ which are a slow-moving copy of $\theta_q$. The overall critic objective is

$$\min_{Q} L = \mathbb{E}_{(s_t,a_t,s_{t+1})\sim\mathcal{D},a_{t+1}^{\pi}\sim\pi(\cdot|f(s_{t+1};\theta_f))} \left[ (y - Q(f(s_t;\theta_f),a_t;\ \theta_q))^2 \right] \tag{2}$$

$$\text{where target value } y = r + \min_{j=1,2}\gamma Q_j(f(s_{t+1};\theta_f),a_{t+1}^{\pi};\ \overline{\theta_q}) \tag{3}$$

### 3.3 Offline Reinforcement Learning

Offline RL problems can be defined as a data-driven formulation of the RL problem. In offline RL, an agent can no longer interact with the environment and collect additional transitions using its learned policy. Instead, the agent was provided with a static dataset of transitions loaded into the replay buffer $\mathcal{D} = \{(s_t^i, a_t^i, r_t^i, s_{t+1}^i)\}_{i=1}^{N}$. In visual offline RL, following DrQv2+BC (Lu et al., 2023), if we consider an additional policy constraint behavioural cloning (BC) term with its strength regulated by a hyperparameter $\lambda_{\text{bc}}$, the policy objective becomes

$$\pi = \arg\max_{\pi}\mathbb{E}_{(s_t,a_t)\sim\mathcal{D},a_t^{\pi}\sim\pi(\cdot|f(s_t;\theta_f))} \left[ Q(f(s_t;\theta_f),a_t^{\pi};\ \theta_q) - \lambda_{\text{bc}}(a_t^{\pi} - a_t)^2 \right] \tag{4}$$

### 3.4 Visual Distractions Dataset

We describe the visual distraction datasets as static datasets of transitions, where observations are sampled from different observation distributions, and the reward and action are sampled from the same distribution. We can define datasets with and without visual distractions as $\mathcal{D}^{\text{normal}} = \{(s_t^{i,\text{normal}}, a_t^{i,\text{normal}}, r_t^{i,\text{normal}}, s_{t+1}^{i,\text{normal}})\}_{i=1}^{N}$ and $\mathcal{D}^{\text{dis}} = \{(s_t^{i,\text{dis}}, a_t^{i,\text{dis}}, r_t^{i,\text{dis}}, s_{t+1}^{i,\text{dis}})\}_{i=1}^{N}$, respectively. The V-D4RL benchmark (Lu et al., 2023) is a more commonly used offline dataset for visual-based offline RL methods but contains only one visual distraction task. The V-D4RL dataset was generated with a fixed visual distraction; that is, the same visual distraction persisted throughout the dataset, whereas during the evaluation, visual distractions were generated randomly.

## 4 Domain Adversarial Training for Visual Distractions

We hypothesise that the subpar performance of the current state-of-the-art baselines can be attributed to visual distractions present during the evaluation but absent from the offline RL training dataset. These unseen visual distractions confuse the encoder, making the encoder estimate less robust latent features $z$. Consequently, the actor-critic backbone, which relies on the latent features $z$, cannot learn robust policy and state-action estimations. Baseline agents can achieve expert-level performance in an environment where the same visual distractions exist in the training dataset. However, when we evaluated the same agent in an environment with random visual distractions, the agent's performance diminished significantly, as shown previously in Figure 2.

In this section, we discuss the proposed approach for improving the robustness of visual-based offline RL for visual distractions. Our proposed method is based on DrQv2+BC (Yarats et al., 2022; Lu et al., 2023) and incorporates two components. We adopt (i) a domain discriminator that trains adversarially against the encoder, and (ii) DropBlock (Ghiasi et al., 2018) layers added to the encoder to achieve robustness against unseen visual distractions. An overview of this architecture is presented in Figure 3.

To address this issue, we propose a framework in which an agent is trained using datasets from two domains: a normal observation domain denoted by $\mathcal{D}^{\text{normal}}$ and a visually distracted domain denoted by $\mathcal{D}^{\text{dis}}$. Intuitively, the presence of domain-specific visual distractions may allow the discriminator to accurately classify latent features belonging to a particular class. However, we aim to induce the opposite and train the visual encoder such that the estimated latent features are indistinguishable from the domain discriminator.

For the actor-critic backbone, we chose the same actor-critic RL backbone as presented in DrQv2 (Yarats et al., 2022; Lu et al., 2023), which is a variant of the actor-critic structure in TD3 (Fujimoto et al., 2018) using a stochastic policy. We use only the distraction latent features $z^{\text{dis}} = f(s^{\text{dis}};\theta_f)$ to train the actor-critic backbone. Only the image encoder, which we hypothesise as the source of underperformance, is trained with both $\mathcal{D}^{\text{normal}}$ and $\mathcal{D}^{\text{dis}}$.

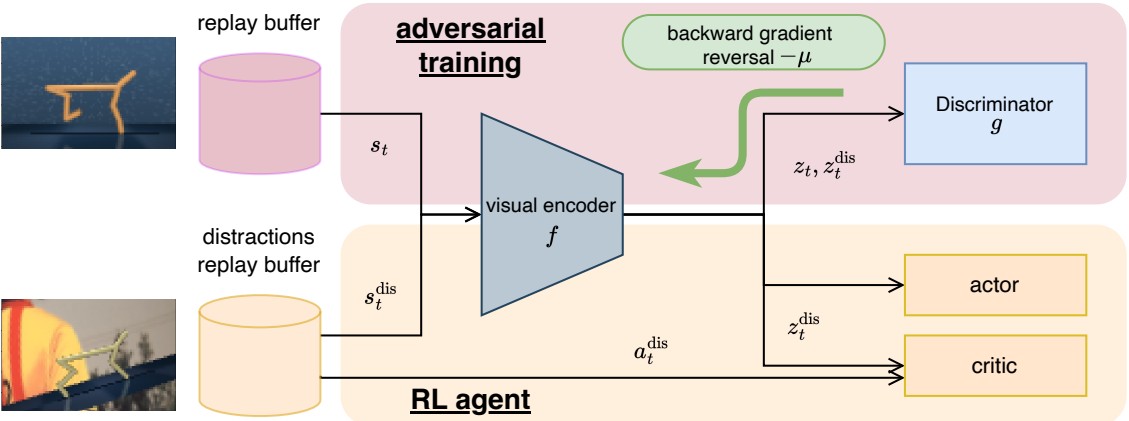

Figure 3: An overview of general visual actor-critic RL methods with our additional proposed discriminator. The lower branch is a common actor-critic backbone where an visual encoder is used to estimate latent proprioceptive features $z$ and uses $z$ to train the RL backbone. The upper adversarial branch is where we reverse the sign of gradients from discriminator loss from the discriminator $g$ such that the encoder is incentivised to learn features that cannot be classified successfully by $g$.

## 4.1 Domain Discriminator

To facilitate domain adversarial learning, we introduce a discriminator $g$ which trains adversarially against the encoder via gradient reversal (Ganin & Lempitsky, 2015). The domain discriminator $g(z; \theta_g)$ maps the latent features $z = f(s; \theta_f)$ with parameters $\theta_g$, and outputs the probability of the latent feature $z$. We use $k$ as the label for classes $k \in \{0, 1\}$, representing normal and distraction observations respectively. Both normal and distraction observations underwent the same data augmentation scheme, which was the same as that in DrQv2 (Yarats et al., 2022) and DrQv2+BC (Lu et al., 2023). Given that we sample $n$ observations from the replay buffers $\mathcal{D}^{\text{normal}}$ and $\mathcal{D}^{\text{dis}}$, the discriminator objective $L_k$ can be expressed as a binary cross-entropy loss, and is defined as follows:

$$L_k = -\frac{1}{n} \sum_{s_i \in (D^{\text{normal}} \cup D^{\text{dis}})} \Big[ k_i \log(g(z_i; \theta_g)) + (1 - k_i) \log(1 - g(z_i; \theta_g)) \Big], \quad \text{where } z = f(s; \theta_f) \qquad (5)$$

Contrary to conventional binary classification tasks, our objective was not to discriminate between different domains. Rather, we aimed to ensure that the discriminator is *unable* to classify encoded latent features when they are similar, irrespective of the domain from which the observations were encoded. To achieve this goal, we adopted the gradient reversal technique proposed for classical domain adaptation tasks (Ganin & Lempitsky, 2015).

Although the discriminator learns to use the encoded latent features $z$ to classify the domains, the encoder learns features that cannot be used to classify the domain successfully because the gradients are reversed. In other words, while the discriminator learns to maximise the latent feature discrepancy between the domains, the encoder attempts to minimise this discrepancy. Consequently, the encoder learns to extract latent features that are shared across both domains, that is, domain-invariant features, while simultaneously suppressing features that are exclusive to one domain. The parameter updates rule for the encoder $f$ and for discriminator $g$ are, in addition to the gradients from the critic objective $L$, as follows, where $\eta_1$ and $\eta_2$ are the learning rates:

$$\theta_f \longleftarrow \quad \theta_f \; - \; \eta_1 \left( \frac{\partial L}{\partial \theta_f} - \mu \frac{\partial L_k}{\partial \theta_f} \right) \qquad \theta_g \longleftarrow \quad \theta_g \; - \; \eta_2 \left( \frac{\partial L_k}{\partial \theta_g} \right) \qquad (6)$$

The encoder parameters $\theta_f$ are jointly updated based on two gradients: one based on the critic objective and the other based on the discriminator objective, as illustrated in Equation (6). The key distinction in a typical update is the $-\mu$ hyperparameter. This hyperparameter is crucial in adversarial training. This $-\mu$ hyperparameter is necessary for preventing a typical encoder update from learning the dissimilarities between features across domains. For all experiments, we selected $\mu = 1$.

## 4.2 DropBlock in Encoder

Common to supervised learning approaches, forcing the encoder not to rely on specific features discourages overfitting the visual distractions found in the offline dataset. To this end, we added DropBlock (Ghiasi et al., 2018) to the visual encoder, driving the encoder to rely on a wider patch of features during the learning process to improve its robustness, thus preventing less robust latent feature $z$ estimations when unseen distractions are present. DropBlock drops patches of a convolutional map (e.g. a $7 \times 7$ map) rather than a single cell (as in dropout (Srivastava et al., 2014)). The visual encoder would be less likely to overfit the detailed observations that are present in the dataset but would gather information that exists in both $\mathcal{D}^{\text{normal}}$ and $\mathcal{D}^{\text{dis}}$, consequently reducing the reliance on background information. We empirically found that adding DropBlock to the visual encoder $f$ frequently enhanced the performance of the proposed domain adversarial training approach. The results are presented in Table 6.

## 5 Novel visual distractions dataset

Table 1: Comparison between V-D4RL distraction benchmark and our novel dataset. ✓ denotes that the data exists. △ denotes that although the distraction data exist for walker-walk random, it was only collected in $64 \times 64$ pixels, whereas the cheetah-run medium-expert data were collected in $84 \times 84$ pixels. In contrast, ours are all $84 \times 84$ pixels.

| Task | Difficulty | V-D4RL distractions | our datasets |
|---|---|---|---|
| cheetah-run | random | | ✓ |
| cheetah-run | medium-expert | ✓ | ✓ |
| ball-in-cup-catch | random | | ✓ |
| ball-in-cup-catch | medium-expert | | ✓ |
| reacher-easy | random | | ✓ |
| reacher-easy | medium-expert | | ✓ |
| reacher-hard | random | | ✓ |
| reacher-hard | medium-expert | | ✓ |
| walker-walk | random | △ | ✓ |
| walker-walk | medium-expert | | ✓ |

V-D4RL (Lu et al., 2023) provides a starting benchmark for visual distractions in offline RL settings. However, specifically for distraction observations, they only collected the *cheetah-run medium-expert* set (in $84 \times 84$-pixel format) and the *walker-walk random* set (in $64 \times 64$-pixel format). Therefore, it is suitable to extend the existing V-D4RL dataset to provide a more comprehensive benchmark for offline continuous control with visual distraction. We collected the Distracting Control Suite (Stone et al., 2021) data and the original DeepMind Control Suite (Tassa et al., 2018) for four additional tasks with three difficulties each, all in a unified $84 \times 84$-pixel format: **medium, expert** and **random** sets (medium and expert are combined to form the medium-expert set). Specifically, we collected for the following tasks: **cheetah-run**, **walker-walk**, **ball-in-cup catch**, **reacher-easy**, and **reacher-hard**. For comparison, we also collected our set for the existing walker-walk random set and the cheetah-run medium-expert set. A brief comparison between the proposed novel dataset and the V-D4RL benchmark is presented in Table 1. For dataset collection, we followed the procedures described in V-D4RL (Lu et al., 2023). A more detailed comparison of the benchmarks is provided in Appendix H.

## 6 Comparison with offline RL algorithms

To evaluate the proposed method, we train four baseline offline RL agents: DrQv2+BC, visual AWAC+BC, and visual IQL and offline-adapted DreamerV2 (Hafner et al., 2021), noted as DV2 (following to V-D4RL). Visual-AWAC+BC is a variant of DrQv2+BC that uses AWAC (Nair et al., 2020) as an actor-critic backbone in addition to the BC term, and visual-IQL uses IQL (Kostrikov et al., 2022) as the backbone (without the BC term). Both use the same visual data augmentation and encoder scheme in DrQv2+BC and only replace the actor-critic method. All four baseline algorithms were trained directly using the distraction dataset. As mentioned in the proposed method, the input domains for the discriminator are normal and dis-x and dis-x $\in$ {easy, medium, hard}. This is to ensure that the RL backbones were trained on the same offline data, and to isolate the effects of the proposed method on the encoder. The input domain for the RL backbone contained distracted data x, x $\in$ {easy, medium, hard}, as noted in the training set column in Table 8 for all algorithms. The discussion and results on two alternative formulations where 1) the encoder of baseline methods were trained on both distracting data, and 2) where our proposed method is trained with two difficulties of distractions (no normal observation) are included in Appendix D. A full list of hyperparameters are shown in Appendix I.4. We follow the original works' hyperparameters for baseline methods. A brief robustness analysis can be found in Appendix G.

All reported numbers were unnormalised, with a maximum reward of 1000 and evaluated over 30 episodes. All reported numbers represent the evaluation results at 1M steps. We trained five seeds: seed $\in$ {0,1,2,3,4}. For the best practices (Agarwal et al., 2021), we report the IQM and 95% stratified bootstrap confidence intervals (CIs). The drop rates for different tasks are presented in Appendix I.

### 6.1 V-D4RL Distraction Dataset

As defined in V-D4RL (Lu et al., 2023) and the Distracting Control Suite (Stone et al., 2021), there are three levels of distractions: distraction-easy, distraction-medium and distraction-hard. The training results evaluated on all three types of distractions (noted respectively as e, m and h), alongside the normal observations (noted as n), are summarised in Table 8. Overall, the experimental results demonstrate that the proposed method consistently outperforms other model-free offline RL baselines in most tasks. DrQv2+BC and IQL were the second-best algorithms overall. Interestingly, training directly on more challenging datasets does not result in better performance, even on difficult tasks, compared with agents trained on simpler datasets and evaluated on more difficult tasks. We hypothesise that this is because the datasets do not contain sufficient variations in distractions for the more difficult distractions; thus, the agents have difficulty learning to deal with them, whereas training on easier tasks allows the agent to infer some domain invariance across distractions.

### 6.2 Our Distraction Dataset

The selected results on the walker-walk medium-expert distraction dataset are shown in Table 3, and an overview of the results is presented in Table 4 for our proposed novel dataset. We observed that our proposed method outperformed all the tasks overall. For more detailed results for our dataset, please refer to Appendix B. The drop rates for different tasks differ; more details regarding the hyperparameters are presented in Appendix I.

## 7 Comparison with visual generalisation methods in RL adapted to offline settings

In this section, we compare the existing generalisation methods in RL that utilise both normal-domain and distracted-domain data. These methods are closely related to our proposed method. The main difference from our work is that these methods were generally developed for online RL with access to two environments, and we assume that we have access to two datasets. Another difference is that these methods often require consecutive training, whereas our method can be trained in an end-to-end manner.

Specifically, we adopted PAD (Hansen et al., 2021b), SVEA (Hansen et al., 2021a), and ILA (Yoneda et al., 2022) for purely offline settings. To facilitate comparison, we adapted these methods to offline RL by initially

Table 2: Training result with V-D4RL cheetah-run medium-expert distraction dataset. The reported numbers are IQM and 95% stratified bootstrap CIs in between parentheses. We mark numbers in bold when one algorithm performs statistically significantly better using t-test with $p \leq 0.05$ compared to other algorithms on IQM. Abbreviations: N=normal, E=easy, M=medium, H=hard.

| train | eval | DrQv2+BC | AWAC+BC | IQL | DV2 | Ours |
|---|---|---|---|---|---|---|
| | | **cheetah-run medium-expert (V-D4RL)** | | | | |
| E | N | 115.1 (51.4, 177.6) | 82.2 (66.0, 250.8) | 113.9 (66.0, 155.6) | 59.6 (21.9, 118.2) | **229.7** (160.6, 251.5) |
| | E | 83.6 (57.9, 96.0) | 60.9 (53.3, 72.8) | 67.1 (54.8, 76.9) | 70.0 (42.2, 92.1) | **130.6** (102.0, 161.0) |
| | M | 35.0 (33.1, 44.5) | 28.5 (20.2, 47.5) | 41.9 (39.7, 54.4) | 76.2 (36.0, 134.6) | **81.0** (66.3, 109.3) |
| | H | 18.1 (14.2, 28.0) | 18.6 (14.8, 22.7) | 20.3 (14.8, 27.6) | **90.2** (42.5, 136.8) | 55.4 (42.2, 60.5) |
| M | N | 69.6 (48.1, 87.2) | 35.1 (3.1, 106.8) | 55.8 (37.6, 115.3) | **84.2** (49.7, 88.5) | **84.4** (36.2, 116.0) |
| | E | 32.5 (22.3, 56.2) | 18.4 (17.5, 26.4) | 39.3 (26.1, 61.5) | 55.3 (41.1, 121.1) | **60.5** (33.0, 72.8) |
| | M | 34.4 (21.3, 49.4) | 24.5 (17.2, 29.8) | 36.9 (29.7, 40.1) | **61.8** (34.8, 88.9) | 49.2 (37.0, 66.9) |
| | H | 13.5 (12.5, 18.6) | 22.0 (17.3, 32.3) | 20.4 (12.7, 24.9) | **65.4** (48.0, 98.8) | 20.2 (13.8, 30.8) |
| H | N | 17.3 (9.9, 29.2) | 9.9 (5.8, 16.0) | 4.9 (2.4, 7.1) | **73.6** (48.6, 134.3) | 39.2 (16.9, 60.8) |
| | E | 14.2 (9.8, 16.5) | 16.6 (13.9, 25.8) | 8.5 (6.0, 12.6) | **76.5** (37.6, 104.7) | 29.4 (21.0, 41.2) |
| | M | 11.6 (10.4, 22.3) | 15.8 (12.5, 21.1) | 10.8 (9.2, 19.5) | **105.5** (82.7, 116.9) | 24.9 (17.7, 28.5) |
| | H | 12.4 (9.1, 15.7) | 16.6 (13.3, 32.7) | 9.8 (9.2, 14.0) | **58.5** (37.6, 81.7) | 27.3 (16.7, 33.6) |
| IQM total | | 457.3 | 349.1 | 430.2 | **876.8** | **831.8** |

Table 3: Results comparison on our dataset walker-walk medium-expert distraction dataset. The reported numbers are IQM and 95% stratified bootstrap CIs in between parentheses. Abbreviations: normal: N, dis-easy: E, dis-medium: M, dis-hard: H.

| train | eval | DrQv2+BC | IQL | DV2 | Ours |
|---|---|---|---|---|---|
| | | **walker-walk medium-expert (ours)** | | | |
| E | N | 469.8 (296.3, 511.3) | 184.7 (37.1, 394.2) | 30.6 (27.3, 45.2) | **633.4** (600.4, 768.2) |
| | E | 231.9 (184.7, 254.0) | 246.4 (165.9, 294.1) | 21.0 (19.4, 28.7) | **293.8** (234.4, 401.3) |
| | M | 118.0 (97.4, 153.4) | 96.7 (70.0, 155.5) | 26.4 (25.2, 31.0) | **156.5** (124.1, 251.7) |
| | H | 70.1 (41.9, 83.5) | 56.2 (40.4, 74.8) | 29.1 (24.5, 30.3) | **104.0** (55.5, 142.2) |
| M | N | 229.8 (121.2, 317.7) | 143.6 (39.5, 345.1) | 31.5 (27.8, 41.8) | **362.9** (145.6, 407.9) |
| | E | 163.8 (113.9, 173.6) | 105.6 (76.1, 170.7) | 32.0 (28.9, 39.7) | **209.1** (178.5, 259.2) |
| | M | 125.2 (108.4, 166.2) | 130.6 (94.7, 140.8) | 35.6 (31.2, 42.1) | **191.1** (164.3, 249.7) |
| | H | 69.0 (58.9, 78.4) | 70.4 (57.6, 96.0) | 31.4 (30.4, 37.3) | **112.5** (104.3, 131.6) |
| H | N | 30.0 (26.5, 65.1) | 30.6 (18.8, 41.9) | **45.1** (35.5, 62.7) | 42.5 (39.2, 81.7) |
| | E | 32.3 (26.1, 37.9) | 31.4 (25.6, 36.5) | **47.5** (35.9, 64.1) | 47.6 (39.4, 80.2) |
| | M | 38.9 (34.3, 42.0) | 31.1 (24.2, 40.6) | 40.2 (34.0, 50.8) | **56.6** (42.5, 79.0) |
| | H | 37.1 (33.3, 71.9) | 30.9 (29.7, 45.6) | 38.4 (30.4, 48.1) | **51.2** (42.0, 71.3) |
| IQM total | | 1615.9 | 1158.2 | 408.8 | **2261.2** |

training them on one distracted observation, then adapting them to the respective test environments, and subsequently demonstrating their performance post-adaptation. All hyperparameters were in accordance with the original authors; we did not make algorithmic changes but only data changes.

In PAD (Hansen et al., 2021b), we used the V-D4RL dataset (distracted observations) as the replay buffer during training and used the inverse dynamics task as the auxiliary training task because it was the best-performing setup in their work. During deployment, we directly adapted to the distracting control suite environment. In the ILA (Yoneda et al., 2022), we train the source domain (in Figure 4 in their paper) purely on the V-D4RL dataset (distracted observations). For the target domain, we adopted the V-D4RL distraction dataset. We used the dynamic model introduced in ILA. For the SVEA, we trained directly on the V-D4RL dataset (distracted observations). The selected results are shown in Table 5, and the full results

Table 4: Aggregated IQM totals on V-D4RL distractions dataset and our novel visual distractions dataset, evaluated on medium, expert and medium-expert difficulty of respective datasets. For some of the tasks below, we only train DrQv2+BC for baseline comparison. Please refer to Appendix B for the full results.

| Task | DrQv2+BC | Ours |
|---|---|---|
| cheetah-run (V-D4RL) | 1407.0 | **2298.0** |
| cheetah-run (ours) | 853.9 | **2035.0** |
| ball-in-cup-catch (ours) | 15298.4 | **18465.9** |
| reacher-easy (ours) | 1792.1 | **2432.6** |
| reacher-hard (ours) | 52.0 | **231.6** |
| walker-walk (ours) | 4644.7 | **6594.3** |

Table 5: Results comparison with related domain adaption methods in RL trained with V-D4RL cheetah-run medium-expert distracting dataset. The reported numbers are IQM and 95% stratified bootstrap CIs in between parentheses.

| | | cheetah-run medium-expert (V-D4RL) | | | |
|---|---|---|---|---|---|
| train | eval | PAD | ILA | SVEA | Ours |
| E | N | 65.8 (43.1, 85.3) | 100.4 (60.7, 124.6) | 39.2 (28.7, 56.8) | **229.7** (160.6, 251.5) |
| | E | 28.7 (24.7, 39.8) | 13.7 (8.7, 28.3) | 28.1 (23.0, 40.8) | **130.6** (102.0, 161.0) |
| | M | 17.5 (13.8, 19.3) | 25.9 (15.6, 31.7) | 36.8 (26.7, 46.4) | **81.0** (66.3, 109.3) |
| | H | 22.4 (16.3, 24.8) | 29.0 (20.1, 40.1) | 57.9 (42.6, 82.0) | **55.4** (42.2, 60.5) |
| M | N | 23.6 (17.6, 26.9) | **155.3** (101.6, 160.7) | 51.2 (38.7, 62.8) | 84.4 (36.2, 116.0) |
| | E | 19.2 (16.4, 28.8) | 60.0 (43.6, 66.8) | 35.0 (25.6, 42.4) | 60.5 (33.0, 72.8) |
| | M | 14.1 (11.9, 19.8) | 34.2 (29.1, 39.5) | 35.8 (28.4, 55.7) | **49.2** (37.0, 66.9) |
| | H | 21.3 (16.7, 26.8) | 26.4 (23.6, 33.0) | **41.2** (25.7, 56.2) | 20.2 (13.8, 30.8) |
| H | N | 16.8 (14.2, 38.2) | **96.4** (89.3, 108.1) | 31.0 (22.6, 37.0) | 39.2 (16.9, 60.8) |
| | E | 16.3 (11.4, 20.8) | **38.3** (22.7, 56.1) | 25.9 (20.1, 28.0) | 29.4 (21.0, 41.2) |
| | M | 18.8 (13.2, 22.0) | **40.5** (23.1, 45.7) | 25.3 (21.4, 32.1) | 24.9 (17.7, 28.5) |
| | H | 20.3 (12.9, 23.3) | 23.4 (22.5, 27.5) | 21.5 (16.0, 34.4) | **27.3** (16.7, 33.6) |
| **IQM total** | | 284.8 | 607.5 | 430.9 | **831.8** |

are summarised in Appendix C. Our proposed method outperforms three domain adaptation approaches in distracted environments.

Interestingly, PAD and SVEA usually exhibit fewer performance drops when trained on an easier distraction dataset and are evaluated in a more difficult environment. Similar to ours, the ILA (Yoneda et al., 2022) uses an adversarial approach, which may explain why our proposed method and the ILA suffer from significant performance drops when distraction difficulties increase, whereas the PAD and SVEA are generally less affected.

## 8 Ablation Studies

### 8.1 DropBlock drop rate

We investigated the effect of DropBlock drop rate using the original V-D4RL cheetah-run distraction dataset. To achieve this, we train agents with $p = \{0.0, 0.1, 0.2, 0.3, 0.5\}$. To understand how the DropBblock helps, we train the proposed adversarial discriminator without a DropBlock ($p = 0.0$) in the enr. We observe that $p = 0.3$ is the best performing overall, whereas having no DropBlock ($p = 0$) is the least performing variation. Selected results is shown in Table 6, and the full results is shown in Table 19.

Table 6: DropBlock ablation results on V-D4RL cheetah-run distraction dataset. The reported numbers are IQM and 95% stratified bootstrap CIs in between parentheses. Abbreviations for dataset type: medium: m, expert: e, medium-expert: m-e. Abbreviations for visual distractions difficulty: normal: N, easy: E, medium: M, hard: H. For example, "e H" equals "expert Hard".

| Task | eval | $p = 0.0$ | $p = 0.1$ | $p = 0.2$ | $p = 0.3$ | $p = 0.5$ |
|------|------|-----------|-----------|-----------|-----------|-----------|
| m-E E | N | 85.4 (45.4, 119.5) | 185.9 (172.9, 243.9) | 171.7 (94.4, 216.1) | **229.7** (160.6, 251.5) | 194.3 (75.8, 208.5) |
|  | E | 71.7 (56.9, 89.9) | **130.6** (102.0, 161.0) | 121.7 (99.9, 141.2) | 110.6 (83.2, 141.2) | 96.8 (91.9, 118.6) |
|  | M | 42.4 (31.9, 48.8) | 68.0 (56.7, 84.9) | **94.0** (57.6, 102.6) | 81.0 (66.3, 109.3) | 77.0 (65.5, 82.0) |
|  | H | 34.3 (32.0, 36.0) | 50.9 (31.3, 53.4) | 49.1 (48.0, 55.4) | **55.4** (42.2, 60.5) | 42.3 (36.7, 46.7) |
| m-E M | N | 63.7 (35.7, 116.4) | 33.5 (14.3, 115.4) | 43.0 (4.5, 108.4) | **84.4** (36.2, 116.0) | 60.2 (26.1, 76.5) |
|  | E | 33.4 (27.8, 47.4) | 27.3 (22.6, 38.2) | 40.5 (31.1, 55.0) | 60.5 (33.0, 72.8) | 53.0 (39.8, 90.6) |
|  | M | 45.0 (31.9, 59.6) | 29.7 (21.0, 32.2) | 31.2 (25.1, 44.0) | 43.9 (22.0, 59.5) | **49.2** (37.0, 66.9) |
|  | H | **25.3** (22.1, 27.5) | 13.2 (10.4, 31.7) | 20.2 (13.8, 30.8) | 13.4 (9.0, 25.7) | 15.3 (12.1, 39.5) |
| m-E H | N | 21.9 (15.7, 39.4) | 6.0 (2.4, 7.7) | 21.5 (17.1, 39.7) | **39.2** (16.9, 60.8) | 24.4 (11.3, 50.0) |
|  | E | 12.0 (10.1, 21.9) | 16.4 (13.0, 21.0) | **29.4** (21.0, 41.2) | 21.5 (14.7, 48.6) | 18.0 (15.6, 23.5) |
|  | M | 14.9 (13.0, 16.0) | 18.6 (11.0, 22.5) | **24.9** (17.7, 28.5) | 19.0 (11.2, 26.2) | 15.0 (11.5, 19.2) |
|  | H | 17.1 (11.5, 23.2) | 18.1 (13.2, 22.4) | **27.3** (16.7, 33.6) | 12.7 (10.9, 22.9) | 12.8 (9.5, 21.0) |
| **IQM total** |  | 467.1 | 598.2 | 674.5 | **771.3** | 658.3 |

## 8.2 Different types of distractions

It is important to understand which types of distraction are easier for the agent to learn, and which types are more difficult. As noted previously, there are three types of distractions: background, viewpoint, and colour, with three levels of distraction respectively. We performed ablation studies on different types by enabling only one distraction at a time and testing at three levels each. The evaluation task is cheetah-run. We used $p = 0.3$ in our proposed algorithm as the basis for ablation. The results for the medium-expert dataset are presented in Table 7, and the full results are presented in Appendix E. Generally, the proposed method performs the best overall. Interestingly, robot colour changes had a minimal effect on the agents' performance. The background changes also had a slight impact. Agents are generally heavily impacted by viewpoint changes, and the final performance is significantly related to the agents' ability to adapt to viewpoint changes.

## 9 Limitations

As mentioned previously, one of the assumption of our proposed method is that two datasets are available. In addition, we also have not discussed in depth the impact of the quality of the two datasets effect performance. For example, one interesting question to ask is, how different must the two datasets be for our proposed method to work? We briefly discuss on using two *distraction* dataset to train our proposed method in Appendix D. An interesting future direction is to further propose novel methods using only distracting data for training.

## 10 Conclusion

In this work, we tackle the issue of visual distraction in offline RL settings, where the distractions that exist in the dataset might significantly differ from those in evaluation. We investigated an adversarial-based algorithm for visual-based offline RL using an additional discriminator with gradient reversal and introduced DropBlock to the visual encoder. Our approach trains the encoder such that it will learn domain-invariant features that are more generalisable when unseen visual distractions are present during evaluation. Our method can be easily implemented by adding a discriminator branch while reversing the gradient flow and adding DropBlock. Despite the simplicity and efficiency of this method, we empirically demonstrated that it achieved excellent performance across various difficulties in the V-D4RL visual distraction dataset, as well

as in our proposed datasets. We demonstrated that our proposed method is more robust to various types of distractions, both individually and in combination. We also showed the importance of having DropBlock to achieve better performance. Additionally, we empirically found that agents are significantly affected by changes in viewpoint, a topic that, to the best of our knowledge, has not been extensively discussed. Future studies should address this issue.

## Acknowledgement

This work was partially supported by JST Moonshot R&D Grant Number JPMJPS2011, CREST Grant Number JPMJCR2015 and Basic Research Grant (Super AI) of Institute for AI and Beyond of the University of Tokyo. This work is also partially supported by JSPS KAKENHI Grant Number JP23K18476.

## Open Source

We have open-sourced our codebase on GitHub with instructions at: `https://github.com/Atine/offlineRL_domain_adversarial`, and our collected dataset can be found in `https://drive.google.com/drive/folders/1J58uGFI2qxTrEJ9LZv4O2iUOUtA3a3lN?usp=sharing`.

Table 7: Distractions ablation results on V-D4RL cheetah-run distraction *medium-expert* dataset, which is essentially an extended version of Table 2. The "all" row is the same as in Table 2. The reported numbers are IQM and 95% stratified bootstrap CIs in between parentheses. Abbreviations for dataset type: medium-expert: m-e. Abbreviations for visual distractions difficulty: normal: N, easy: E, medium: M, hard: H. The ablation studies for the expert and medium distractions datasets are presented in Appendix E.

| train | eval | type of dis | DrQv2+BC | AWAC+BC | IQL | Ours |
|---|---|---|---|---|---|---|
| m-e E | E | background | 91.3 (50.8, 111.8) | 69.1 (66.0, 103.8) | 97.8 (71.1, 112.0) | **215.2** (187.3, 220.5) |
| | E | viewpoint | 47.1 (23.0, 74.1) | 51.3 (28.2, 119.6) | 106.3 (50.4, 120.1) | **170.7** (145.5, 192.3) |
| | E | colour | 69.0 (45.4, 140.7) | 95.6 (46.5, 143.8) | 114.0 (75.5, 158.1) | **227.9** (202.1, 269.1) |
| | E | all | 83.6 (57.9, 96.0) | 60.9 (53.3, 72.8) | 67.1 (54.8, 76.9) | **130.6** (102.0, 161.0) |
| | M | background | 111.4 (102.2, 123.1) | 95.8 (80.3, 128.5) | 157.3 (95.7, 188.6) | **236.8** (199.6, 244.9) |
| | M | viewpoint | 32.9 (14.2, 44.7) | 28.3 (16.7, 68.6) | 48.2 (20.6, 59.2) | **94.3** (81.5, 119.9) |
| | M | colour | 95.0 (53.2, 146.8) | 98.2 (49.3, 143.2) | 96.0 (68.8, 142.0) | **217.0** (189.0, 282.3) |
| | M | all | 35.0 (33.1, 44.5) | 28.5 (20.2, 47.5) | 41.9 (39.7, 54.4) | **81.0** (66.3, 109.3) |
| | H | background | 92.7 (78.6, 160.3) | 71.1 (61.3, 104.5) | 107.7 (91.5, 155.1) | **194.7** (149.4, 222.3) |
| | H | viewpoint | 21.0 (10.5, 25.3) | 16.7 (12.2, 30.4) | 25.4 (17.1, 41.7) | **65.3** (47.3, 74.8) |
| | H | colour | 71.2 (51.5, 122.1) | 105.8 (50.8, 166.7) | 98.8 (59.5, 125.2) | **200.2** (162.4, 239.5) |
| | H | all | 18.1 (14.2, 28.0) | 18.6 (14.8, 22.7) | 20.3 (14.8, 27.6) | **55.4** (42.2, 60.5) |
| m-e M | E | background | 25.0 (14.3, 47.6) | 27.6 (21.5, 79.9) | 18.2 (12.9, 51.7) | **49.3** (33.5, 63.6) |
| | E | viewpoint | 37.2 (27.1, 41.6) | 53.9 (22.3, 85.0) | 29.3 (19.2, 79.3) | **80.8** (51.8, 102.7) |
| | E | colour | 55.3 (38.7, 74.6) | 49.9 (19.2, 84.6) | 53.9 (32.0, 72.3) | **70.8** (38.7, 90.5) |
| | E | all | 32.5 (22.3, 56.2) | 18.4 (17.5, 26.4) | 39.3 (26.1, 61.5) | **60.5** (33.0, 72.8) |
| | M | background | 44.0 (28.6, 64.0) | 40.2 (35.0, 85.6) | 25.3 (20.9, 69.7) | **56.0** (47.6, 73.1) |
| | M | viewpoint | 23.5 (12.5, 30.4) | **48.1** (21.8, 70.0) | 17.1 (13.8, 35.1) | 44.7 (28.6, 61.1) |
| | M | colour | 39.2 (24.3, 50.7) | 57.3 (26.8, 72.7) | 44.3 (32.5, 64.5) | **76.2** (38.3, 99.9) |
| | M | all | 34.4 (21.3, 49.4) | 24.5 (17.2, 29.8) | 36.9 (29.7, 40.1) | **49.2** (37.0, 66.9) |
| | H | background | 39.4 (29.8, 51.7) | 41.7 (31.7, 51.8) | 34.6 (28.1, 69.0) | **68.5** (33.2, 86.4) |
| | H | viewpoint | 15.5 (10.6, 18.3) | 24.5 (15.7, 32.0) | 16.4 (9.4, 23.5) | **35.3** (17.9, 53.8) |
| | H | colour | 40.6 (25.9, 66.1) | 50.4 (31.5, 71.3) | 41.7 (25.7, 66.1) | **84.0** (42.8, 96.9) |
| | H | all | 13.5 (12.5, 18.6) | **22.0** (17.3, 32.3) | 20.4 (12.7, 24.9) | 20.2 (13.8, 30.8) |
| m-e H | E | background | 11.6 (9.5, 32.6) | 22.2 (19.0, 34.0) | 6.7 (5.1, 12.8) | **44.3** (27.1, 79.8) |
| | E | viewpoint | 16.4 (9.0, 36.8) | 9.2 (7.2, 14.3) | 7.7 (6.6, 10.3) | **32.4** (22.6, 46.5) |
| | E | colour | 16.9 (9.0, 28.0) | 11.2 (9.6, 21.7) | 6.3 (3.8, 11.3) | **43.8** (19.4, 64.4) |
| | E | all | 14.2 (9.8, 16.5) | 16.6 (13.9, 25.8) | 8.5 (6.0, 12.6) | **29.4** (21.0, 41.2) |
| | M | background | 18.0 (12.2, 21.0) | 22.9 (15.5, 34.3) | 8.6 (6.7, 13.1) | **34.0** (26.3, 78.5) |
| | M | viewpoint | 14.8 (7.2, 24.8) | 12.7 (10.6, 15.2) | 9.2 (6.7, 14.6) | **27.1** (21.2, 45.3) |
| | M | colour | 14.5 (9.8, 24.1) | 18.0 (11.7, 24.2) | 10.7 (7.0, 16.3) | **41.9** (18.3, 54.8) |
| | M | all | 11.6 (10.4, 22.3) | 15.8 (12.5, 21.1) | 10.8 (9.2, 19.5) | **24.9** (17.7, 28.5) |
| | H | background | 14.0 (11.3, 20.4) | 20.5 (18.4, 28.1) | 8.2 (8.1, 9.3) | **27.9** (20.3, 45.5) |
| | H | viewpoint | 10.8 (7.0, 20.7) | 12.3 (11.0, 17.9) | 8.6 (6.7, 16.7) | **32.7** (22.5, 36.3) |
| | H | colour | 15.9 (9.7, 25.7) | 13.6 (10.3, 23.8) | 13.1 (7.9, 16.7) | **29.1** (17.4, 36.2) |
| | H | all | 12.4 (9.1, 15.7) | 16.6 (13.3, 32.7) | 9.8 (9.2, 14.0) | **27.3** (16.7, 33.6) |

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

## A    Visualisations of tasks

Brief visualisation of tasks is shown below in Figure 4 with different difficulties of the tasks.

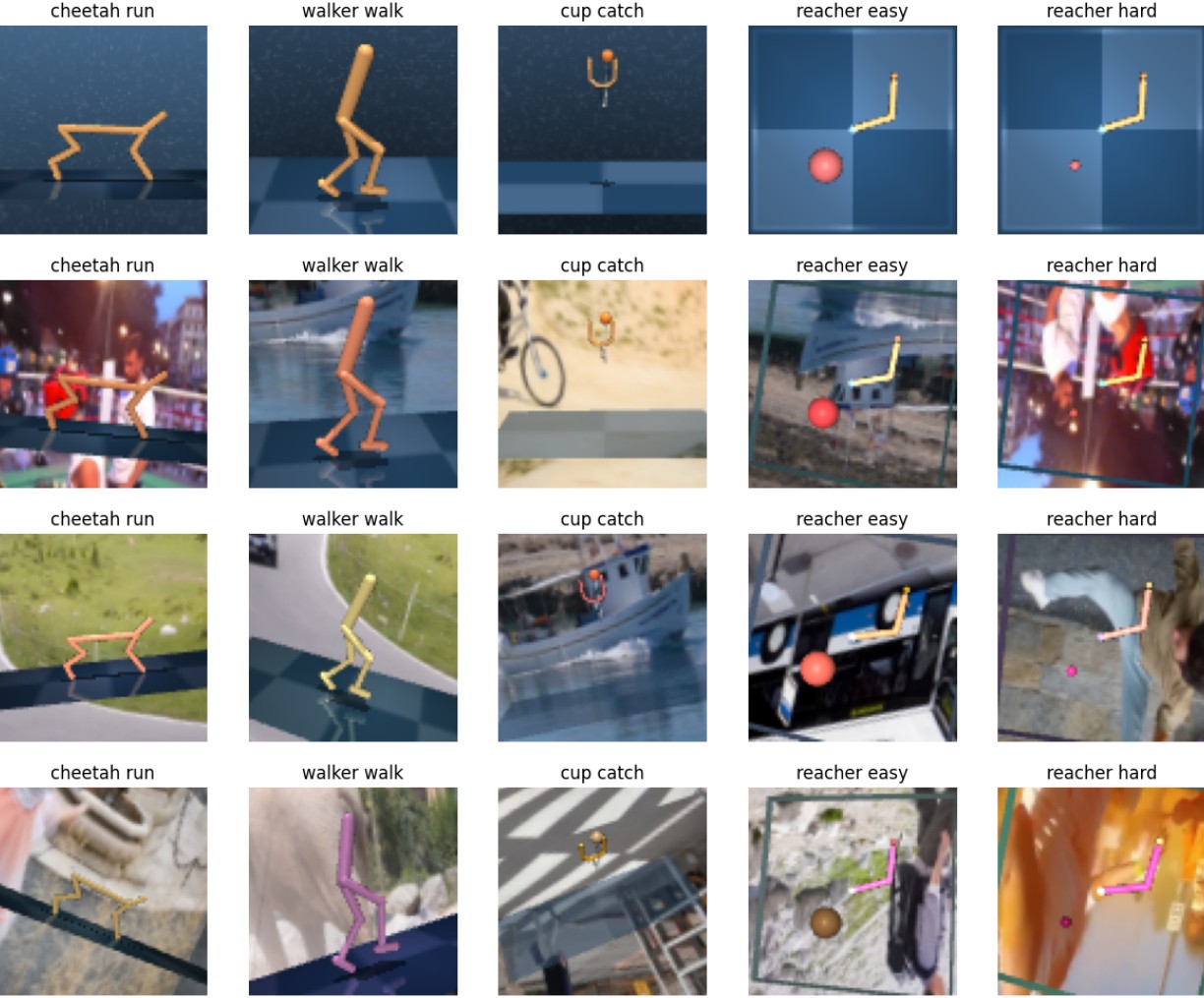

Figure 4: Brief visualisation of tasks showing the different difficulties of the tasks. From top to bottom are normal observations, dis-easy observations, dis-medium observations and dis-hard observations.

## B   Full experiment results with offline RL algorithms

In this section, we include the full table results train on V-D4RL cheetah-run distraction dataset, our proposed cheetah-run distraction dataset, ball-in-cup-catch distraction dataset, reacher-easy distraction dataset, reacher-hard distraction dataset and walker-walk distraction dataset. For V-D4RL dataset, we compare with baseline agents DrQv2+BC, AWAC+BC and IQL. For our datasets, we mainly compared with DrQV2+BC as it was the best performing for the V-D4RL dataset, with the exception of walker-walk distraction dataset where we included IQL to compare.

Table 8: Training result with V-D4RL cheetah-run distraction dataset. The reported numbers are IQM and 95% stratified bootstrap CIs in between parentheses. We mark numbers in bold when one algorithm performs statistically significantly better using t-test with $p \leq 0.05$ compared to other algorithms on IQM. Abbreviations: N: normal, E: dis-easy, M: dis-medium, H=dis-hard. e: expert. m: medium. m-e: medium-expert.

| train | eval | cheetah-run (V-D4RL) | | | | |
|---|---|---|---|---|---|---|
| | | DrQv2+BC | AWAC+BC | IQL | DV2 | Ours |
| e E | N | 108.7 (42.0, 132.9) | 44.6 (26.3, 129.5) | 59.1 (27.4, 97.3) | 124.2 (92.1, 193.3) | **233.7** (197.9, 256.8) |
| | E | 34.2 (30.5, 47.3) | 31.4 (18.4, 45.2) | 50.0 (41.9, 95.6) | 61.6 (48.1, 105.1) | **100.8** (83.6, 113.7) |
| | M | 13.7 (11.6, 21.7) | 7.8 (5.4, 25.1) | 17.3 (7.6, 22.1) | **141.2** (130.4, 150.1) | 42.0 (35.6, 73.5) |
| | H | 9.7 (7.4, 12.7) | 6.6 (4.5, 8.0) | 6.8 (5.0, 14.7) | **61.0** (48.4, 154.7) | 33.6 (22.3, 39.1) |
| e M | N | 18.9 (4.2, 64.7) | 41.1 (11.7, 93.3) | **252.8** (197.0, 346.7) | 145.4 (76.6, 233.1) | 76.5 (28.2, 110.3) |
| | E | 33.3 (18.5, 40.7) | 15.0 (9.3, 21.2) | 15.2 (5.9, 28.8) | **106.4** (80.0, 133.0) | 31.1 (26.0, 51.1) |
| | M | 28.3 (22.7, 39.3) | 17.9 (8.8, 23.1) | 11.4 (5.6, 13.9) | **85.8** (60.4, 186.1) | 27.7 (24.7, 41.5) |
| | H | 11.1 (9.3, 14.8) | 7.1 (5.7, 13.1) | 5.3 (4.5, 8.8) | **145.3** (86.7, 196.8) | 20.1 (15.9, 27.3) |
| e H | N | 1.5 (1.0, 3.4) | 5.0 (1.7, 15.3) | 11.2 (4.1, 24.7) | **79.5** (58.1, 163.0) | 24.3 (17.8, 44.8) |
| | E | 9.1 (8.4, 14.2) | 14.4 (11.9, 16.8) | 8.3 (6.2, 12.6) | **69.6** (52.6, 83.9) | 18.8 (12.6, 22.2) |
| | M | 9.8 (7.9, 17.8) | 12.4 (10.2, 13.9) | 11.5 (8.1, 15.5) | **76.0** (35.3, 123.2) | 16.2 (12.9, 18.8) |
| | H | 11.1 (7.8, 17.2) | 13.6 (10.1, 19.0) | 8.7 (6.0, 11.0) | **75.6** (53.5, 95.2) | 12.4 (9.8, 18.5) |
| m E | N | 131.0 (115.0, 143.2) | 79.6 (31.3, 145.2) | 124.9 (52.2, 176.6) | 186.1 (130.7, 208.4) | **246.9** (219.7, 279.7) |
| | E | 99.5 (65.6, 110.4) | 56.4 (44.0, 77.7) | 98.8 (63.1, 118.4) | 83.8 (66.6, 117.1) | **152.0** (122.0, 178.9) |
| | M | 41.4 (32.4, 56.8) | 23.2 (17.2, 42.8) | 26.1 (20.0, 70.4) | **182.6** (101.6, 228.5) | 68.2 (49.0, 83.3) |
| | H | 18.6 (12.7, 22.6) | 14.4 (8.7, 19.5) | 11.2 (7.6, 18.6) | **128.9** (94.1, 171.1) | 34.0 (26.9, 41.5) |
| m M | N | 125.9 (81.9, 145.0) | 70.4 (49.9, 133.2) | 28.3 (15.4, 72.2) | 128.7 (73.9, 179.2) | 116.4 (68.0, 154.3) |
| | E | 37.6 (31.0, 53.2) | 18.6 (11.9, 26.8) | 27.0 (17.1, 41.6) | **63.3** (41.9, 80.5) | 45.5 (36.0, 97.9) |
| | M | 40.3 (23.6, 52.5) | 14.3 (8.7, 17.7) | 15.6 (13.5, 19.8) | **131.4** (82.5, 189.3) | 50.3 (27.5, 52.9) |
| | H | 21.2 (14.4, 29.6) | 9.8 (7.2, 20.6) | 8.1 (7.2, 29.1) | **71.8** (62.2, 74.4) | 26.5 (19.7, 38.0) |
| m H | N | 60.3 (43.5, 70.7) | 19.3 (4.2, 77.9) | 13.6 (6.3, 24.7) | **194.9** (101.6, 233.5) | 36.2 (27.6, 47.7) |
| | E | 26.3 (19.3, 33.6) | 21.8 (13.0, 31.3) | 8.7 (5.5, 14.5) | **112.4** (86.7, 218.1) | 18.5 (16.0, 24.8) |
| | M | 24.5 (20.4, 32.6) | 14.0 (7.1, 21.6) | 10.6 (7.6, 16.2) | **172.6** (113.7, 176.8) | 18.3 (14.4, 21.6) |
| | H | 33.7 (21.1, 37.6) | 12.8 (9.3, 16.2) | 15.6 (11.7, 21.2) | **123.6** (90.1, 207.7) | 16.2 (15.6, 19.9) |
| m-e E | N | 115.1 (51.4, 177.6) | 82.2 (66.0, 250.8) | 113.9 (66.0, 155.6) | 59.6 (19.5, 118.2) | **229.7** (160.6, 251.5) |
| | E | 83.6 (57.9, 96.0) | 60.9 (53.3, 72.8) | 67.1 (54.8, 76.9) | 70.0 (43.8, 92.2) | **130.6** (102.0, 161.0) |
| | M | 35.0 (33.1, 44.5) | 28.5 (20.2, 47.5) | 41.9 (39.7, 54.4) | 76.2 (36.5, 136.9) | **81.0** (66.3, 109.3) |
| | H | 18.1 (14.2, 28.0) | 18.6 (14.8, 22.7) | 20.3 (14.8, 27.6) | **90.2** (42.4, 136.8) | 55.4 (42.2, 60.5) |
| m-e M | N | 69.6 (48.1, 87.2) | 35.1 (3.1, 106.8) | 55.8 (37.6, 115.3) | 84.2 (50.3, 88.6) | 84.4 (36.2, 116.0) |
| | E | 32.5 (22.3, 56.2) | 18.4 (17.5, 26.4) | 39.3 (26.1, 61.5) | 55.3 (40.6, 112.3) | **60.5** (33.0, 72.8) |
| | M | 34.4 (21.3, 49.4) | 24.5 (17.2, 29.8) | 36.9 (29.7, 40.1) | **61.8** (34.8, 89.2) | 49.2 (37.0, 66.9) |
| | H | 13.5 (12.5, 18.6) | 22.0 (17.3, 32.3) | 20.4 (12.7, 24.9) | **65.4** (48.0, 98.7) | 20.2 (13.8, 30.8) |
| m-e H | N | 17.3 (9.9, 29.2) | 9.9 (5.8, 16.0) | 4.9 (2.4, 7.1) | **73.6** (48.6, 134.3) | 39.2 (16.9, 60.8) |
| | E | 14.2 (9.8, 16.5) | 16.6 (13.9, 25.8) | 8.5 (6.0, 12.6) | **76.5** (36.2, 104.7) | 29.4 (21.0, 41.2) |
| | M | 11.6 (10.4, 22.3) | 15.8 (12.5, 21.1) | 10.8 (9.2, 19.5) | **105.5** (82.7, 116.9) | 24.9 (17.7, 28.5) |
| | H | 12.4 (9.1, 15.7) | 16.6 (13.3, 32.7) | 9.8 (9.2, 14.0) | **58.5** (37.9, 81.7) | 27.3 (16.7, 33.6) |
| IQM total | | 1407.0 | 920.6 | 1275.7 | **3628.5** | 2298.0 |

Table 9: Results comparison on our proposed walker-walk distraction dataset. The reported numbers are IQM and 95% stratified bootstrap CIs in between parentheses. Abbreviations: normal: N, dis-easy: E, dis-medium: M, dis-hard: H. e: expert. m: medium. m-e: medium-expert.

| | | walker-walk (ours) | | | |
|---|---|---|---|---|---|
| train | eval | DrQv2+BC | IQL | DV2 | Ours |
| E | N | 532.8 (503.7, 564.4) | 244.3 (92.6, 423.8) | 30.0 (25.4, 40.9) | **633.2** (609.8, 691.9) |
| | E | 190.5 (145.3, 217.5) | 186.0 (154.7, 211.7) | 41.2 (27.5, 51.7) | **289.5** (269.8, 398.5) |
| | M | 82.5 (71.9, 117.7) | 76.9 (56.5, 92.8) | 26.8 (21.3, 32.9) | **137.9** (129.2, 234.3) |
| | H | 63.5 (52.2, 73.0) | 51.5 (37.2, 65.0) | 28.8 (22.8, 69.3) | **96.9** (67.0, 121.2) |
| M | N | 91.1 (47.3, 154.8) | 59.1 (28.9, 92.8) | 27.3 (25.7, 54.0) | **246.4** (183.5, 276.0) |
| | E | 99.6 (94.4, 113.8) | 50.7 (45.2, 59.1) | 31.7 (27.4, 44.5) | **170.6** (110.9, 213.9) |
| | M | 96.5 (65.4, 104.5) | 51.6 (46.1, 91.9) | 24.0 (21.2, 27.0) | **154.9** (145.1, 205.5) |
| | H | 65.6 (46.2, 74.0) | 50.3 (45.4, 58.2) | 31.6 (30.5, 54.1) | **79.9** (65.8, 89.8) |
| H | N | **48.3** (24.8, 65.6) | 22.4 (17.5, 29.1) | 38.1 (33.6, 42.8) | 38.9 (33.6, 59.1) |
| | E | **44.3** (26.2, 46.2) | 29.7 (27.2, 33.6) | 42.0 (33.2, 48.0) | 36.8 (35.2, 63.8) |
| | M | 38.3 (32.7, 43.5) | 28.8 (26.7, 32.3) | 33.4 (25.6, 43.6) | 40.6 (35.3, 52.8) |
| | H | 43.5 (32.3, 47.0) | 35.5 (32.1, 43.5) | 26.3 (24.0, 57.0) | **48.5** (42.3, 63.8) |
| E | N | 531.0 (469.1, 555.2) | 312.8 (111.1, 476.1) | 30.0 (25.3, 40.9) | 544.9 (533.5, 555.1) |
| | E | 156.6 (138.0, 176.6) | 205.1 (134.3, 251.2) | 41.2 (27.5, 51.7) | **279.1** (257.1, 347.0) |
| | M | 81.8 (68.9, 115.9) | 96.0 (63.2, 108.9) | 26.8 (21.4, 32.9) | **160.1** (132.4, 203.8) |
| | H | 66.7 (50.4, 83.9) | 49.0 (42.9, 64.4) | 28.8 (23.0, 69.3) | **73.0** (58.5, 146.6) |
| M | N | 247.9 (113.9, 325.3) | 104.0 (56.5, 168.5) | 27.3 (25.7, 53.4) | **420.8** (336.5, 488.0) |
| | E | 175.6 (127.2, 249.7) | 119.9 (106.4, 212.5) | 31.7 (27.4, 44.5) | **263.6** (201.4, 317.1) |
| | M | 125.8 (102.7, 148.3) | 77.7 (56.7, 102.1) | 24.0 (21.2, 27.0) | **227.9** (155.8, 258.1) |
| | H | 92.1 (83.2, 98.7) | 85.6 (66.9, 96.5) | 31.6 (30.5, 53.8) | **117.4** (84.5, 178.4) |
| H | N | 27.8 (20.3, 60.8) | 32.8 (24.1, 53.4) | 38.1 (33.6, 42.8) | **71.2** (54.6, 132.1) |
| | E | 36.5 (31.9, 40.2) | 35.6 (24.1, 42.7) | 42.0 (34.1, 48.0) | **64.4** (48.8, 90.7) |
| | M | 49.9 (40.7, 59.9) | 39.2 (27.2, 45.9) | 33.4 (25.6, 43.6) | **68.6** (46.2, 96.4) |
| | H | 40.6 (36.3, 45.3) | 39.8 (34.6, 44.3) | 26.3 (24.0, 57.1) | **68.0** (63.1, 82.8) |
| E | N | 469.8 (296.3, 511.3) | 184.7 (37.1, 394.2) | 30.6 (27.3, 45.2) | **633.4** (600.4, 768.2) |
| | E | 231.9 (184.7, 254.0) | 246.4 (165.9, 294.1) | 21.0 (19.4, 28.7) | **293.8** (234.4, 401.3) |
| | M | 118.0 (97.4, 153.4) | 96.7 (70.0, 155.5) | 26.4 (25.2, 31.0) | **156.5** (124.1, 251.7) |
| | H | 70.1 (41.9, 83.5) | 56.2 (40.4, 74.8) | 29.1 (24.5, 30.3) | **104.0** (55.5, 142.2) |
| M | N | 229.8 (121.2, 317.7) | 143.6 (39.5, 345.1) | 31.5 (27.8, 41.8) | **362.9** (145.6, 407.9) |
| | E | 163.8 (113.9, 173.6) | 105.6 (76.1, 170.7) | 32.0 (28.9, 39.7) | **209.1** (178.5, 259.2) |
| | M | 125.2 (108.4, 166.2) | 130.6 (94.7, 140.8) | 35.6 (31.2, 42.1) | **191.1** (164.3, 249.7) |
| | H | 69.0 (58.9, 78.4) | 70.4 (57.6, 96.0) | 31.4 (30.4, 37.3) | **112.5** (104.3, 131.6) |
| H | N | 30.0 (26.5, 65.1) | 30.6 (18.8, 41.9) | **45.1** (35.5, 62.7) | 42.5 (39.2, 81.7) |
| | E | 32.3 (26.1, 37.9) | 31.4 (25.6, 36.5) | 47.5 (35.9, 64.1) | 47.6 (39.4, 80.2) |
| | M | 38.9 (34.3, 42.0) | 31.1 (24.2, 40.6) | 40.2 (34.0, 50.8) | **56.6** (42.5, 79.0) |
| | H | 37.1 (33.3, 71.9) | 30.9 (29.7, 45.6) | 38.4 (30.4, 48.1) | **51.2** (42.0, 71.3) |
| IQM total | | 4644.7 | 4518.2 | 1204.4 | **6594.3** |

Table 10: Results comparison on our proposed cheetah-run distraction dataset. The reported numbers are IQM and 95% stratified bootstrap CIs in between parentheses. Abbreviations: normal: N, dis-easy: E, dis-medium: M, dis-hard: H. e: expert. m: medium. m-e: medium-expert.

| | | cheetah-run (ours) | |
|---|---|---|---|
| train | eval | DrQv2+BC | Ours |
| e E | N | 8.4 (2.5, 16.5) | **114.9** (40.3, 133.8) |
| | E | 22.4 (12.2, 35.5) | **54.6** (50.9, 81.2) |
| | M | 12.7 (8.6, 17.4) | **39.8** (31.4, 44.5) |
| | H | 5.4 (4.3, 6.2) | **21.9** (15.7, 30.1) |
| e M | N | 9.9 (5.8, 18.8) | **22.0** (7.9, 47.0) |
| | E | 5.1 (3.8, 7.7) | **24.2** (21.8, 27.5) |
| | M | 9.3 (6.0, 14.0) | **23.0** (15.3, 32.0) |
| | H | 6.1 (4.7, 7.3) | **9.8** (8.1, 11.7) |
| e H | N | 3.5 (1.7, 4.3) | **17.0** (11.7, 122.1) |
| | E | 7.0 (5.3, 9.9) | **13.4** (8.9, 17.1) |
| | M | 8.2 (7.7, 10.4) | 9.3 (6.1, 14.7) |
| | H | **10.6** (6.7, 15.1) | 7.2 (6.0, 18.0) |
| m E | N | 132.3 (46.4, 165.9) | **205.6** (105.3, 295.7) |
| | E | 90.5 (50.4, 95.4) | **149.3** (106.1, 168.3) |
| | M | 35.3 (28.6, 50.8) | **63.9** (53.3, 81.2) |
| | H | 19.1 (17.3, 21.1) | **41.6** (33.6, 52.2) |
| m M | N | 46.1 (16.8, 103.9) | **163.2** (144.4, 197.9) |
| | E | 18.4 (12.6, 26.0) | **72.6** (42.0, 77.7) |
| | M | 28.5 (15.5, 42.5) | **68.5** (60.8, 73.4) |
| | H | 13.8 (10.6, 24.1) | **37.3** (30.0, 53.1) |
| m H | N | 12.3 (6.4, 16.3) | **25.8** (14.4, 47.3) |
| | E | 11.8 (9.5, 16.7) | **23.3** (17.0, 42.2) |
| | M | 16.7 (10.4, 18.5) | **22.1** (13.6, 28.9) |
| | H | 18.7 (12.4, 24.4) | **23.4** (15.7, 35.2) |
| m-e E | N | 47.3 (18.8, 78.6) | **212.4** (185.9, 231.4) |
| | E | 61.6 (42.7, 65.0) | **151.3** (108.2, 178.5) |
| | M | 46.0 (30.0, 60.9) | **91.8** (80.0, 101.3) |
| | H | 26.1 (19.9, 37.0) | **58.2** (34.7, 76.6) |
| m-e M | N | 24.1 (11.6, 69.6) | **67.0** (39.1, 191.0) |
| | E | 27.7 (16.4, 37.1) | **48.0** (37.7, 66.4) |
| | M | 23.2 (14.9, 27.6) | **34.3** (18.7, 56.4) |
| | H | 15.0 (12.5, 16.9) | **26.3** (16.6, 41.9) |
| m-e H | N | 2.6 (1.3, 4.0) | **37.2** (19.2, 82.7) |
| | E | 9.5 (7.2, 14.7) | **20.0** (13.5, 29.3) |
| | M | 8.2 (6.5, 10.2) | **17.4** (12.2, 26.6) |
| | H | 10.5 (8.9, 11.6) | **17.4** (14.0, 24.6) |
| total | | 853.9 | **2035.0** |

Table 11: Results comparison on our proposed ball-in-cup-catch distraction dataset. The reported numbers are IQM and 95% stratified bootstrap CIs in between parentheses. Abbreviations: normal: N, dis-easy: E, dis-medium: M, dis-hard: H. e: expert. m: medium. m-e: medium-expert.

| | | ball-in-cup-catch (ours) | |
|---|---|---|---|
| train | eval | DrQv2+BC | Ours |
| e E | N | 329.7 (0.0, 797.8) | **641.7** (189.8, 974.4) |
| | E | 329.7 (0.0, 797.8) | **640.6** (189.4, 973.1) |
| | M | 329.7 (0.0, 797.8) | **633.5** (185.2, 967.3) |
| | H | 329.7 (0.0, 797.8) | **639.8** (189.0, 972.6) |
| e M | N | 330.3 (0.0, 798.2) | **659.3** (197.6, 991.6) |
| | E | 330.3 (0.0, 800.0) | **557.5** (163.7, 896.8) |
| | M | 330.5 (0.0, 798.3) | **581.0** (153.2, 938.3) |
| | H | 330.3 (0.0, 800.0) | **560.8** (148.5, 882.6) |
| e H | N | 330.0 (0.0, 796.2) | **656.0** (195.6, 987.6) |
| | E | 330.0 (0.0, 796.2) | **650.5** (194.6, 982.6) |
| | M | 341.0 (13.2, 802.8) | **581.1** (154.5, 935.8) |
| | H | 339.3 (5.6, 800.4) | **596.5** (165.4, 927.4) |
| m E | N | 968.3 (918.8, 985.2) | 972.0 (966.0, 992.8) |
| | E | 541.0 (308.0, 856.1) | 578.4 (334.4, 884.5) |
| | M | 369.0 (111.8, 791.9) | **465.8** (213.7, 825.9) |
| | H | 300.2 (81.9, 759.2) | **400.7** (108.1, 796.1) |
| m M | N | **965.3** (365.0, 980.4) | 662.0 (198.2, 995.4) |
| | E | 452.6 (193.7, 831.2) | 430.8 (139.6, 817.5) |
| | M | 371.2 (152.3, 824.1) | **452.4** (196.7, 843.3) |
| | H | 398.7 (127.9, 779.4) | 375.0 (154.9, 778.2) |
| m H | N | 484.7 (119.6, 833.2) | **595.3** (159.4, 952.8) |
| | E | **289.7** (192.5, 827.4) | 242.5 (117.4, 785.3) |
| | M | 224.1 (80.6, 766.8) | 241.3 (75.3, 704.8) |
| | H | 245.2 (69.2, 682.0) | **296.0** (46.9, 807.5) |
| m-e E | N | 958.3 (191.6, 980.6) | 950.0 (825.6, 980.6) |
| | E | 432.1 (167.4, 829.2) | **574.1** (271.9, 896.9) |
| | M | 362.7 (88.0, 801.6) | **442.9** (159.2, 830.2) |
| | H | 313.6 (67.6, 809.1) | **413.7** (117.7, 814.3) |
| m-e M | N | **972.7** (192.6, 984.6) | 563.7 (143.4, 926.6) |
| | E | **506.0** (214.7, 832.4) | 366.8 (57.4, 803.4) |
| | M | 355.6 (103.5, 790.5) | 399.0 (110.7, 789.1) |
| | H | 362.0 (60.0, 800.0) | 344.2 (52.2, 792.2) |
| m-e H | N | **633.3** (218.8, 804.2) | 332.3 (0.0, 799.4) |
| | E | 203.9 (70.4, 610.9) | **322.2** (59.9, 794.6) |
| | M | 289.3 (33.5, 675.1) | 331.0 (17.3, 790.4) |
| | H | 318.4 (46.0, 719.0) | 315.5 (17.1, 766.1) |
| IQM total | | 15298.4 | **18465.9** |

Table 12: Results comparison on our proposed reacher-easy distraction dataset. The reported numbers are IQM and 95% stratified bootstrap CIs in between parentheses. Abbreviations: normal: N, dis-easy: E, dis-medium: M, dis-hard: H. e: expert. m: medium. m-e: medium-expert.

| | | reacher-easy (ours) | |
|---|---|---|---|
| train | eval | DrQv2+BC | Ours |
| e E | N | 48.3 (23.6, 551.2) | 46.0 (6.6, 213.6) |
| | E | **61.6** (30.9, 91.1) | 52.5 (12.2, 212.9) |
| | M | **62.2** (31.6, 120.2) | 49.7 (8.8, 213.6) |
| | H | **61.3** (31.4, 100.6) | 48.6 (7.5, 213.6) |
| e M | N | **85.7** (25.8, 173.4) | 46.3 (7.8, 214.4) |
| | E | **71.7** (29.8, 108.2) | 45.4 (7.8, 220.4) |
| | M | **79.8** (34.4, 132.4) | 44.8 (7.8, 218.7) |
| | H | **68.4** (27.8, 111.1) | 44.7 (7.8, 217.9) |
| e H | N | 41.3 (14.2, 112.5) | 46.7 (7.8, 211.2) |
| | E | 43.7 (21.0, 56.5) | 46.2 (5.2, 212.8) |
| | M | 58.6 (26.6, 83.5) | 46.7 (5.2, 209.7) |
| | H | 48.8 (32.7, 114.5) | 46.7 (5.2, 209.7) |
| m E | N | 14.6 (7.6, 760.0) | **315.3** (0.0, 771.2) |
| | E | 132.2 (70.1, 463.4) | **199.8** (83.0, 362.4) |
| | M | 45.2 (29.6, 358.3) | **80.6** (62.9, 375.8) |
| | H | 57.7 (26.3, 197.0) | **87.5** (48.0, 265.7) |
| m M | N | 30.0 (13.0, 661.8) | **129.6** (0.0, 748.8) |
| | E | **28.7** (16.2, 266.0) | 12.3 (6.0, 457.4) |
| | M | 37.6 (21.7, 312.9) | **102.5** (40.3, 325.3) |
| | H | 11.7 (6.8, 310.3) | **45.6** (13.7, 262.7) |
| m H | N | 0.0 (0.0, 232.0) | 4.3 (2.0, 755.0) |
| | E | 24.9 (14.5, 271.4) | **91.2** (40.8, 207.0) |
| | M | 37.5 (17.2, 259.1) | **49.2** (29.0, 239.9) |
| | H | 18.6 (4.6, 332.0) | **36.5** (19.0, 230.8) |
| m-e E | N | **62.3** (18.2, 130.4) | 50.4 (26.2, 648.3) |
| | E | 43.6 (24.7, 292.4) | **84.0** (29.0, 183.4) |
| | M | **63.0** (26.7, 216.3) | 49.3 (26.7, 186.0) |
| | H | **59.6** (24.5, 218.8) | 38.3 (21.7, 179.6) |
| m-e M | N | **69.1** (32.1, 763.0) | 47.9 (16.7, 64.1) |
| | E | 54.7 (20.7, 288.2) | **78.5** (30.6, 109.3) |
| | M | 28.0 (17.8, 205.4) | **86.9** (48.8, 139.1) |
| | H | 68.5 (43.0, 237.6) | 66.2 (36.8, 159.5) |
| m-e H | N | 0.0 (0.0, 752.8) | **24.3** (7.8, 723.8) |
| | E | **93.1** (40.7, 338.4) | 69.5 (41.0, 251.3) |
| | M | **42.0** (27.5, 310.6) | 21.1 (15.7, 301.8) |
| | H | 38.1 (29.4, 196.8) | **57.1** (33.3, 295.3) |
| **IQM total** | | 1792.1 | **2432.6** |

Table 13: Results comparison on our proposed reacher-hard distraction dataset. The reported numbers are IQM and 95% stratified bootstrap CIs in between parentheses. Abbreviations: normal: N, dis-easy: E, dis-medium: M, dis-hard: H. e: expert. m: medium. m-e: medium-expert.

| | | reacher-hard (ours) | |
|---|---|---|---|
| train | eval | DrQv2+BC | Ours |
| e E | N | 0.0 (0.0, 4.0) | 1.0 (0.0, 3.0) |
| | E | 0.0 (0.0, 4.0) | 1.0 (0.0, 3.2) |
| | M | 0.0 (0.0, 4.0) | 1.0 (0.0, 3.2) |
| | H | 0.0 (0.0, 4.0) | 1.0 (0.0, 3.0) |
| e M | N | 0.0 (0.0, 3.2) | 1.7 (0.0, 4.0) |
| | E | 0.0 (0.0, 3.2) | 1.7 (0.0, 4.0) |
| | M | 0.0 (0.0, 3.2) | 1.7 (0.0, 4.0) |
| | H | 0.0 (0.0, 3.2) | 1.7 (0.0, 4.0) |
| e H | N | 0.0 (0.0, 4.0) | 1.3 (0.0, 6.4) |
| | E | 0.0 (0.0, 4.1) | 1.2 (0.0, 6.3) |
| | M | 0.0 (0.0, 4.0) | 1.3 (0.0, 6.3) |
| | H | 0.0 (0.0, 4.0) | 1.3 (0.0, 6.1) |
| m E | N | 0.7 (0.4, 20.3) | 2.3 (0.0, 8.6) |
| | E | 1.0 (0.9, 32.6) | **30.2** (13.6, 52.1) |
| | M | 1.1 (0.5, 2.3) | **8.1** (0.5, 27.0) |
| | H | 1.4 (1.1, 4.9) | 2.5 (0.6, 30.5) |
| m M | N | 0.0 (0.0, 0.0) | 0.0 (0.0, 33.0) |
| | E | 0.3 (0.1, 0.9) | 0.5 (0.3, 1.2) |
| | M | 0.4 (0.3, 1.0) | **15.2** (5.1, 54.1) |
| | H | **12.0** (1.1, 27.0) | 0.8 (0.5, 4.9) |
| m H | N | 0.0 (0.0, 0.0) | 0.0 (0.0, 3.2) |
| | E | 1.4 (0.7, 2.3) | 2.7 (0.9, 8.0) |
| | M | 1.5 (1.1, 2.4) | 0.6 (0.3, 29.9) |
| | H | 0.9 (0.6, 1.4) | **32.1** (13.1, 50.6) |
| m-e E | N | 0.1 (0.0, 1.6) | 0.0 (0.0, 20.9) |
| | E | 3.1 (1.1, 5.5) | **8.6** (4.7, 28.2) |
| | M | 0.7 (0.3, 5.3) | **7.0** (0.3, 26.6) |
| | H | 4.1 (3.1, 21.8) | **16.6** (5.8, 26.3) |
| m-e M | N | 0.4 (0.3, 35.2) | 0.0 (0.0, 42.4) |
| | E | 3.6 (2.1, 6.6) | **10.7** (4.1, 36.7) |
| | M | 3.9 (1.9, 5.6) | **10.0** (4.8, 16.0) |
| | H | 5.8 (2.5, 6.9) | **12.4** (7.6, 23.7) |
| m-e H | N | 0.0 (0.0, 0.0) | **25.3** (0.0, 638.8) |
| | E | 3.6 (2.4, 6.7) | **11.0** (2.4, 42.1) |
| | M | 1.5 (1.4, 9.4) | **11.7** (1.5, 29.5) |
| | H | 4.5 (2.7, 25.9) | **7.4** (3.3, 17.0) |
| total | | 52.0 | **231.6** |

## C   Full experiment results with visual generalisation methods in RL adapted to offline settings

Table 14: Results comparison with related domain adaption methods in RL. The reported numbers are IQM and 95% stratified bootstrap CIs in between parentheses.

| train | eval | cheetah-run (V-D4RL) PAD | ILA | SVEA | Ours |
|---|---|---|---|---|---|
| e E | N | 55.3 (33.9, 67.8) | 133.8 (104.5, 158.6) | 32.8 (29.0, 38.4) | **233.7** (197.9, 256.8) |
| | E | 25.7 (20.5, 27.2) | 45.2 (41.4, 55.7) | 25.5 (18.4, 35.5) | **100.8** (83.6, 113.7) |
| | M | 14.7 (13.4, 15.9) | **53.0** (37.7, 64.3) | 29.0 (23.8, 37.3) | 42.0 (35.6, 73.5) |
| | H | 18.4 (16.4, 19.3) | 32.3 (30.0, 38.7) | **46.9** (40.8, 49.9) | 33.6 (22.3, 39.1) |
| e M | N | 24.8 (13.7, 33.6) | **115.3** (87.9, 141.6) | 37.7 (36.1, 54.2) | 76.5 (28.2, 110.3) |
| | E | 17.2 (13.2, 22.2) | **54.9** (45.5, 57.5) | 42.5 (34.8, 47.9) | 31.1 (26.0, 51.1) |
| | M | 13.9 (10.0, 15.7) | 30.9 (28.1, 41.9) | **39.6** (35.2, 48.6) | 27.7 (24.7, 41.5) |
| | H | 18.3 (12.1, 20.6) | 30.1 (26.8, 33.9) | **40.2** (28.5, 47.6) | 20.1 (15.9, 27.3) |
| e H | N | 17.4 (13.8, 26.9) | **99.1** (87.4, 120.2) | 42.3 (18.4, 48.3) | 24.3 (17.8, 44.8) |
| | E | 14.6 (12.5, 15.7) | **29.1** (17.9, 47.7) | 39.1 (21.9, 47.8) | 18.8 (12.6, 22.2) |
| | M | 16.9 (13.2, 20.4) | **39.2** (14.3, 43.9) | 37.9 (17.5, 47.9) | 16.2 (12.9, 18.8) |
| | H | 16.2 (14.3, 19.1) | 23.5 (10.6, 25.6) | **49.6** (15.2, 52.1) | 12.4 (9.8, 18.5) |
| m E | N | 23.3 (18.4, 30.5) | 102.5 (90.1, 221.3) | 21.5 (13.0, 24.5) | **246.9** (219.7, 279.7) |
| | E | 17.8 (14.6, 23.3) | 37.1 (29.6, 48.0) | 11.2 (5.6, 17.5) | **152.0** (122.0, 178.9) |
| | M | 11.1 (9.2, 13.4) | 42.4 (28.9, 56.7) | 12.6 (5.4, 17.6) | **68.2** (49.0, 83.3) |
| | H | 15.4 (11.9, 19.6) | 34.7 (28.7, 40.5) | **45.1** (30.9, 61.3) | 34.0 (26.9, 41.5) |
| m M | N | 27.2 (21.5, 33.3) | 93.5 (80.1, 105.1) | 32.5 (25.5, 34.2) | 116.4 (68.0, 154.3) |
| | E | 12.2 (9.5, 15.8) | **55.0** (47.9, 61.4) | 21.5 (16.2, 23.1) | 45.5 (36.0, 97.9) |
| | M | 9.2 (8.0, 12.9) | 41.7 1.0, 45.5 | 23.1 (18.1, 26.0) | **50.3** (27.5, 52.9) |
| | H | 11.0 (9.0, 16.2) | 17.7 12.2, 21.4 | **39.8** (27.7, 48.4) | 26.5 (19.7, 38.0) |
| m H | N | 12.5 (8.0, 17.6) | **109.9** (81.4, 143.5) | 27.6 (21.5, 31.4) | 36.2 (27.6, 47.7) |
| | E | 13.6 (10.5, 17.2) | **46.9** (39.0, 53.5) | 25.8 (17.7, 28.3) | 18.5 (16.0, 24.8) |
| | M | 16.2 (13.9, 19.4) | **33.1** (24.1, 47.0) | 20.4 (17.1, 23.8) | 18.3 (14.4, 21.6) |
| | H | 15.6 (13.0, 18.0) | **24.3** (20.6, 37.9) | 18.2 (10.0, 25.9) | 16.2 (15.6, 19.9) |
| m-e E | N | 65.8 (43.1, 85.3) | 100.4 (60.7, 124.6) | 39.2 (28.7, 56.8) | **229.7** (160.6, 251.5) |
| | E | 28.7 (24.7, 39.8) | 13.7 (8.7, 28.3) | 28.1 (23.0, 40.8) | **130.6** (102.0, 161.0) |
| | M | 17.5 (13.8, 19.3) | 25.9 (15.6, 31.7) | 36.8 (26.7, 46.4) | **81.0** (66.3, 109.3) |
| | H | 22.4 (16.3, 24.8) | 29.0 (20.1, 40.1) | 57.9 (42.6, 82.0) | **55.4** (42.2, 60.5) |
| m-e M | N | 23.6 (17.6, 26.9) | **155.3** (101.6, 160.7) | 51.2 (38.7, 62.8) | 84.4 (36.2, 116.0) |
| | E | 19.2 (16.4, 28.8) | 60.0 (43.6, 66.8) | 35.0 (25.6, 42.4) | 60.5 (33.0, 72.8) |
| | M | 14.1 (11.9, 19.8) | 34.2 (29.1, 39.5) | 35.8 (28.4, 55.7) | **49.2** (37.0, 66.9) |
| | H | 21.3 (16.7, 26.8) | 26.4 (23.6, 33.0) | 41.2 (25.7, 56.2) | 20.2 (13.8, 30.8) |
| m-e H | N | 16.8 (14.2, 38.2) | **96.4** (89.3, 108.1) | 31.0 (22.6, 37.0) | 39.2 (16.9, 60.8) |
| | E | 16.3 (11.4, 20.8) | **38.3** (22.7, 56.1) | 25.9 (20.1, 28.0) | 29.4 (21.0, 41.2) |
| | M | 18.8 (13.2, 22.0) | **40.5** (23.1, 45.7) | 25.3 (21.4, 32.1) | 24.9 (17.7, 28.5) |
| | H | 20.3 (12.9, 23.3) | 23.4 (22.5, 27.5) | 21.5 (16.0, 34.4) | **27.3** (16.7, 33.6) |
| total | | 723.3 | 1968.7 | 1191.3 | **2298.0** |

## D    Additional experiments on Data Usage

In the section, we show the results of an alternative formulation where the encoder of the baseline algotihm uses both normal dataset and distraction dataset during training. DrQv2+BC denotes the baseline algorithm, DrQv2+BC, both data denotes the baseline algorithm where its encoder has access to both distracting data and normal data, and Ours denotes our proposed algorithm, as shown in Table 15. Furthermore, we can train another version where our proposed algorithm is trained on two distracting datasets, shown in Table 16.

Table 15: Results comparison on cheetah-run distraction dataset (V-D4RL). The reported numbers are IQM and 95% stratified bootstrap CIs in between parentheses. Abbreviations: normal: N, dis-easy: E, dis-medium: M, dis-hard: H. e: expert. m: medium. m-e: medium-expert. Our method performs better when the quality of data is mixed (medium-expert), while DrQv2+BC, both data performs much better on normal observation.

| train | eval | cheetah-run (V-D4RL) | | |
| --- | --- | --- | --- | --- |
| | | **DrQv2+BC** | **DrQv2+BC, both data** | **Ours** |
| e E | N | 108.7 (42.0, 132.9) | **579.2** (542.4, 661.9) | 233.7 (197.9, 256.8) |
| | E | 34.2 (30.5, 47.3) | 56.6 (52.2, 73.9) | **100.8** (83.6, 113.7) |
| | M | 13.7 (11.6, 21.7) | 23.3 (19.0, 28.0) | **42.0** (35.6, 73.5) |
| | H | 9.7 (7.4, 12.7) | 12.2 (10.4, 14.0) | **33.6** (22.3, 39.1) |
| e M | N | 18.9 (4.2, 64.7) | **569.0** (432.5, 603.0) | 76.5 (28.2, 110.3) |
| | E | 33.3 (18.5, 40.7) | 36.3 (29.7, 44.8) | 31.1 (26.0, 51.1) |
| | M | 28.3 (22.7, 39.3) | 27.0 (16.9, 36.4) | 27.7 (24.7, 41.5) |
| | H | 11.1 (9.3, 14.8) | 12.7 (8.6, 18.0) | **20.1** (15.9, 27.3) |
| e H | N | 1.5 (1.0, 3.4) | **419.0** (148.2, 480.0) | 24.3 (17.8, 44.8) |
| | E | 9.1 (8.4, 14.2) | **21.8** (12.5, 40.2) | 18.8 (12.6, 22.2) |
| | M | 9.8 (7.9, 17.8) | **21.9** (14.4, 29.8) | 16.2 (12.9, 18.8) |
| | H | 11.1 (7.8, 17.2) | **16.6** (13.4, 27.5) | 12.4 (9.8, 18.5) |
| m E | N | 131.0 (115.0, 143.2) | **562.4** (555.5, 577.5) | 246.9 (219.7, 279.7) |
| | E | 99.5 (65.6, 110.4) | 96.5 (78.6, 106.2) | **152.0** (122.0, 178.9) |
| | M | 41.4 (32.4, 56.8) | 66.5 (40.4, 70.6) | **68.2** (49.0, 83.3) |
| | H | 18.6 (12.7, 22.6) | 25.4 (19.3, 28.4) | **34.0** (26.9, 41.5) |
| m M | N | 125.9 (81.9, 145.0) | **548.5** (541.4, 560.3) | 116.4 (68.0, 154.3) |
| | E | 37.6 (31.0, 53.2) | **63.2** (53.1, 91.3) | 45.5 (36.0, 97.9) |
| | M | 40.3 (23.6, 52.5) | 49.7 (37.6, 59.1) | 50.3 (27.5, 52.9) |
| | H | 21.2 (14.4, 29.6) | 23.0 (17.2, 37.1) | **26.5** (19.7, 38.0) |
| m H | N | 60.3 (43.5, 70.7) | **531.6** (515.1, 549.5) | 36.2 (27.6, 47.7) |
| | E | 26.3 (19.3, 33.6) | **65.2** (45.7, 79.1) | 18.5 (16.0, 24.8) |
| | M | 24.5 (20.4, 32.6) | **40.9** (35.2, 65.5) | 18.3 (14.4, 21.6) |
| | H | **33.7** (21.1, 37.6) | 26.4 (21.0, 36.5) | 16.2 (15.6, 19.9) |
| m-e E | N | 115.1 (51.4, 177.6) | 93.3 (35.5, 142.8) | **229.7** (160.6, 251.5) |
| | E | 83.6 (57.9, 96.0) | 69.6 (53.8, 129.7) | **130.6** (102.0, 161.0) |
| | M | 35.0 (33.1, 44.5) | 28.2 (20.8, 38.8) | **81.0** (66.3, 109.3) |
| | H | 18.1 (14.2, 28.0) | 20.6 (16.0, 29.5) | **55.4** (42.2, 60.5) |
| m-e M | N | 69.6 (48.1, 87.2) | 88.2 (35.6, 137.2) | 84.4 (36.2, 116.0) |
| | E | 32.5 (22.3, 56.2) | 33.1 (27.0, 54.0) | **60.5** (33.0, 72.8) |
| | M | 34.4 (21.3, 49.4) | 23.5 (20.7, 26.1) | **49.2** (37.0, 66.9) |
| | H | 13.5 (12.5, 18.6) | 13.5 (11.9, 29.3) | **20.2** (13.8, 30.8) |
| m-e H | N | 17.3 (9.9, 29.2) | 7.7 (3.0, 18.2) | **39.2** (16.9, 60.8) |
| | E | 14.2 (9.8, 16.5) | 14.5 (11.3, 19.2) | **29.4** (21.0, 41.2) |
| | M | 11.6 (10.4, 22.3) | 16.3 (14.4, 20.6) | **24.9** (17.7, 28.5) |
| | H | 12.4 (9.1, 15.7) | 15.1 (10.8, 19.4) | **27.3** (16.7, 33.6) |
| **IQM total** | | 1407.0 | **4318.5** | 2298.0 |

Table 16: Results comparison on cheetah-run distraction dataset (V-D4RL). The reported numbers are IQM and 95% stratified bootstrap CIs in between parentheses. Abbreviations: normal: N, dis-easy: E, dis-medium: M, dis-hard: H. e: expert. m: medium. m-e: medium-expert. Two distraction data is trained on **easy distraction and medium distraction**, with dropblock drop rate $p = 0.3$. Without tuning the dropblock probability $p$, we can observe that even when using two different difficulties of distraction observations data for training, our proposed method can achieve similar performance compared to training on normal and distraction observation.

| | | cheetah-run (V-D4RL) | | |
|---|---|---|---|---|
| train | eval | Ours, easy+medium (p=0.3) | Ours, normal+dis(p=0.3) | Ours normal+dis (best p) |
| e E | N | 202.6 (170.8, 231.6) | 233.7 (197.9, 256.8) | 233.7 (197.9, 256.8) |
| | E | 85.3 (63.6, 99.9) | 88.3 (72.1, 107.4) | **100.8** (83.6, 113.7) |
| | M | 44.0 (27.0, 48.2) | 44.9 (34.3, 57.1) | 42.0 (35.6, 73.5) |
| | H | 30.9 (26.9, 42.9) | 21.4 (17.3, 28.1) | **33.6** (22.3, 39.1) |
| e M | N | 78.5 (58.6, 102.3) | 76.5 (28.2, 110.3) | 76.5 (28.2, 110.3) |
| | E | 30.3 (25.0, 36.8) | 32.8 (29.2, 37.4) | 31.1 (26.0, 51.1) |
| | M | 29.6 (28.7, 40.6) | 23.0 (17.4, 27.4) | 27.7 (24.7, 41.5) |
| | H | 16.3 (11.6, 23.9) | 20.1 (9.7, 24.0) | **20.1** (15.9, 27.3) |
| e H | N | 13.4 (7.9, 15.2) | 24.3 (17.8, 44.8) | 24.3 (17.8, 44.8) |
| | E | 7.7 (5.8, 11.6) | 16.0 (12.8, 21.0) | 18.8 (12.6, 22.2) |
| | M | 11.9 (10.4, 18.0) | 16.2 (12.9, 18.8) | 16.2 (12.9, 18.8) |
| | H | 10.8 (8.3, 12.8) | 9.8 (7.5, 11.9) | 12.4 (9.8, 18.5) |
| m E | N | 222.1 (123.8, 318.5) | 246.9 (219.7, 279.7) | 246.9 (219.7, 279.7) |
| | E | 129.1 (106.7, 142.6) | 152.0 (122.0, 178.9) | **152.0** (122.0, 178.9) |
| | M | 70.8 (60.2, 83.0) | 64.0 (52.2, 77.8) | **68.2** (49.0, 83.3) |
| | H | 38.4 (30.4, 50.1) | 30.7 (24.6, 35.7) | **34.0** (26.9, 41.5) |
| m M | N | 86.3 (73.5, 121.8) | 116.4 (68.0, 154.3) | 116.4 (68.0, 154.3) |
| | E | **52.8** (43.3, 59.5) | 45.5 (36.0, 97.9) | 45.5 (36.0, 97.9) |
| | M | 40.0 (31.1, 46.9) | 30.7 (21.1, 40.4) | **50.3** (27.5, 52.9) |
| | H | 21.0 (18.1, 28.5) | 23.9 (15.4, 37.4) | **26.5** (19.7, 38.0) |
| m H | N | 24.3 (11.3, 40.0) | 36.2 (27.6, 47.7) | 36.2 (27.6, 47.7) |
| | E | 8.4 (6.5, 25.7) | 14.8 (8.7, 21.3) | 18.5 (16.0, 24.8) |
| | M | 8.7 (6.5, 23.2) | 12.1 (7.6, 14.2) | 18.3 (14.4, 21.6) |
| | H | 7.9 (4.6, 11.2) | 11.4 (7.7, 20.4) | 16.2 (15.6, 19.9) |
| m-e E | N | 218.5 (194.5, 235.2) | 229.7 (160.6, 251.5) | **229.7** (160.6, 251.5) |
| | E | 145.0 (129.7, 178.9) | 110.6 (83.2, 141.2) | **130.6** (102.0, 161.0) |
| | M | 81.9 (68.1, 95.9) | 81.0 (66.3, 109.3) | **81.0** (66.3, 109.3) |
| | H | 50.2 (39.5, 61.4) | 55.4 (42.2, 60.5) | **55.4** (42.2, 60.5) |
| m-e M | N | **119.5** (99.8, 142.2) | 84.4 (36.2, 116.0) | 84.4 (36.2, 116.0) |
| | E | 45.0 (36.1, 60.4) | 60.5 (33.0, 72.8) | **60.5** (33.0, 72.8) |
| | M | 39.1 (32.4, 51.2) | 43.9 (22.0, 59.5) | **49.2** (37.0, 66.9) |
| | H | **20.5** (18.0, 30.1) | 13.4 (9.0, 25.7) | **20.2** (13.8, 30.8) |
| m-e H | N | 19.5 (7.8, 29.3) | 39.2 (16.9, 60.8) | **39.2** (16.9, 60.8) |
| | E | **29.5** (19.5, 42.8) | 21.5 (14.7, 48.6) | **29.4** (21.0, 41.2) |
| | M | 16.3 (8.9, 29.0) | 19.0 (11.2, 26.2) | **24.9** (17.7, 28.5) |
| | H | 24.0 (12.4, 28.8) | 12.7 (10.9, 22.9) | **27.3** (16.7, 33.6) |
| **IQM total** | | 2080.1 | 2162.9 | 2298.0 |

# E  Ablation on Distractions

Table 17: Distractions ablation results on V-D4RL cheetah-run distraction *expert* benchmark. The reported numbers are IQM and 95% stratified bootstrap CIs in between parentheses. Abbreviations for dataset type: expert: e. Abbreviations for visual distractions difficulty: normal: N, easy: E, medium: M, hard: H.

| train | eval | type of dis | DrQv2+BC | AWAC+BC | IQL | Ours |
|---|---|---|---|---|---|---|
| e E | E | background | 37.9 (22.8, 58.1) | 49.8 (29.6, 72.7) | 40.1 (25.4, 147.7) | 135.6 (110.5, 244.6) |
| e E | E | viewpoint | 67.1 (26.4, 91.2) | 32.8 (20.4, 64.9) | 45.3 (23.9, 80.3) | 137.0 (101.5, 150.4) |
| e E | E | colour | 102.3 (36.0, 127.3) | 49.0 (27.2, 127.7) | 67.7 (40.0, 95.9) | 203.0 (157.8, 226.3) |
| e E | E | all | 34.2 (30.5, 47.3) | 31.4 (18.4, 45.2) | 50.0 (41.9, 95.6) | 100.8 (83.6, 113.7) |
| e E | M | background | 43.8 (25.6, 59.3) | 37.1 (24.1, 58.8) | 69.8 (34.8, 114.1) | 173.2 (128.1, 212.2) |
| e E | M | viewpoint | 34.7 (16.1, 44.8) | 17.2 (9.7, 21.0) | 17.6 (9.8, 22.3) | 60.3 (47.4, 72.1) |
| e E | M | colour | 98.1 (42.0, 121.6) | 55.6 (36.6, 118.4) | 61.8 (47.0, 69.9) | 170.9 (130.7, 207.3) |
| e E | M | all | 13.7 (11.6, 21.7) | 7.8 (5.4, 25.1) | 17.3 (7.6, 22.1) | 42.0 (35.6, 73.5) |
| e E | H | background | 55.6 (30.1, 68.6) | 20.6 (12.75, 34.2) | 35.2 (16.2, 68.4) | 127.3 (97.8, 162.9) |
| e E | H | viewpoint | 12.8 (9.5, 17.7) | 12.4 (7.79, 18.1) | 9.9 (6.31, 17.1) | 46.9 (32.9, 53.9) |
| e E | H | colour | 70.1 (33.4, 95.6) | 52.1 (36.89, 73.9) | 53.9 (32.07, 61.7) | 170.8 (137.6, 187.4) |
| e E | H | all | 9.7 (7.4, 12.7) | 6.6 (4.5, 8.0) | 6.8 (5.0, 14.7) | 33.6 (22.3, 39.1) |
| e M | E | background | 23.7 (13.5, 49.3) | 8.4 (7.5, 19.1) | 43.0 (20.0, 91.6) | 29.6 (16.4, 38.7) |
| e M | E | viewpoint | 20.8 (9.4, 36.7) | 25.3 (6.0, 69.2) | 118.5 (85.9, 127.9) | 95.8 (28.1, 107.0) |
| e M | E | colour | 22.6 (13.0, 42.2) | 25.8 (6.8, 52.1) | 319.8 (272.1, 359.1) | 84.5 (32.2, 105.6) |
| e M | E | all | 33.3 (18.5, 40.7) | 15.0 (9.3, 21.2) | 15.2 (5.9, 28.8) | 31.1 (26.0, 51.1) |
| e M | M | background | 45.6 (31.6, 61.3) | 23.9 (11.5, 44.3) | 31.2 (15.5, 67.1) | 27.9 (18.7, 35.3) |
| e M | M | viewpoint | 23.3 (13.0, 36.9) | 14.0 (3.7, 35.4) | 57.2 (44.7, 69.8) | 62.1 (31.7, 83.1) |
| e M | M | colour | 31.8 (11.9, 36.5) | 28.2 (7.0, 57.5) | 251.4 (184.8, 267.2) | 77.5 (41.9, 94.6) |
| e M | M | all | 28.3 (22.7, 39.3) | 17.9 (8.8, 23.1) | 11.4 (5.6, 13.9) | 27.7 (24.7, 41.5) |
| e M | H | background | 31.1 (19.2, 40.5) | 20.7 (15.2, 23.5) | 23.7 (21.6, 41.8) | 23.7 (19.7, 29.0) |
| e M | H | viewpoint | 12.5 (7.4, 19.6) | 6.4 (4.4, 35.0) | 24.8 (13.9, 29.4) | 36.6 (21.4, 39.9) |
| e M | H | colour | 36.2 (20.1, 42.1) | 29.2 (10.3, 72.6) | 200.6 (192.8, 221.3) | 65.1 (42.4, 91.0) |
| e M | H | all | 11.1 (9.3, 14.8) | 7.1 (5.7, 13.1) | 5.3 (4.5, 8.8) | 20.1 (15.9, 27.3) |
| e H | E | background | 12.4 (9.4, 22.1) | 14.0 (11.0, 26.1) | 11.6 (10.6, 18.7) | 20.6 (17.8, 28.7) |
| e H | E | viewpoint | 4.1 (2.4, 10.0) | 3.1 (2.0, 9.7) | 8.9 (3.6, 11.9) | 16.9 (13.2, 18.8) |
| e H | E | colour | 4.4 (2.6, 9.1) | 3.9 (2.2, 8.5) | 11.8 (7.4, 20.4) | 20.6 (15.5, 39.2) |
| e H | E | all | 9.1 (8.4, 14.2) | 14.4 (11.9, 16.8) | 8.3 (6.2, 12.6) | 18.8 (12.6, 22.2) |
| e H | M | background | 12.2 (9.0, 29.6) | 22.2 (14.5, 29.1) | 9.9 (7.1, 11.5) | 25.8 (17.9, 54.9) |
| e H | M | viewpoint | 5.9 (3.4, 7.7) | 3.9 (2.6, 9.0) | 5.0 (2.6, 7.6) | 12.6 (11.2, 22.7) |
| e H | M | colour | 5.5 (2.7, 8.2) | 5.3 (2.8, 8.1) | 10.9 (7.4, 21.4) | 24.5 (16.6, 32.2) |
| e H | M | all | 9.8 (7.9, 17.8) | 12.4 (10.2, 13.9) | 11.5 (8.1, 15.5) | 16.2 (12.9, 18.8) |
| e H | H | background | 12.4 (10.8, 20.1) | 13.1 (10.2, 17.7) | 11.3 (7.8, 16.3) | 22.6 (16.7, 27.1) |
| e H | H | viewpoint | 5.6 (3.5, 9.4) | 4.6 (3.5, 8.5) | 5.9 (3.3, 16.8) | 21.3 (14.7, 28.1) |
| e H | H | colour | 7.0 (5.2, 10.5) | 5.6 (3.3, 8.1) | 10.7 (7.5, 23.0) | 18.5 (15.3, 26.7) |
| e H | H | all | 11.1 (7.8, 17.2) | 13.6 (10.1, 19.0) | 8.7 (6.0, 11.0) | 12.4 (9.8, 18.5) |

Table 18: Distractions ablation results on V-D4RL cheetah-run distraction *medium* benchmark. The reported numbers are IQM and 95% stratified bootstrap CIs in between parentheses. Abbreviations for dataset type: medium: m. Abbreviations for visual distractions difficulty: normal: N, easy: E, medium: M, hard: H.

| train | eval | type of dis | DrQv2+BC | AWAC+BC | IQL | Ours |
|-------|------|-------------|----------|---------|-----|------|
| m E | E | background | 129.7 (115.1, 163.5) | 101.5 (62.6, 123.7) | 165.4 (82.7, 167.5) | 293.1 (258.5, 317.0) |
| m E | E | viewpoint | 69.9 (61.1, 103.1) | 72.1 (42.7, 105.3) | 59.8 (26.8, 99.0) | 148.1 (119.4, 172.9) |
| m E | E | colour | 125.7 (115.88, 136.8) | 84.0 (35.9, 130.1) | 117.5 (44.3, 166.7) | 223.8 (208.1, 301.9) |
| m E | E | all | 99.5 (65.6, 110.4) | 56.4 (44.0, 77.7) | 98.8 (63.1, 118.4) | 152.0 (122.0, 178.9) |
| m E | M | background | 53.9 (35.6, 66.6) | 8.8 (7.2, 30.0) | 10.9 (2.6, 30.5) | 30.3 (23.9, 70.3) |
| m E | M | viewpoint | 105.6 (85.0, 120.0) | 68.4 (49.8, 83.9) | 33.4 (21.4, 52.3) | 98.7 (62.0, 117.7) |
| m E | M | colour | 95.6 (90.1, 103.6) | 76.4 (62.2, 90.0) | 45.4 (32.1, 92.7) | 103.1 (63.5, 121.2) |
| m E | M | all | 41.4 (32.4, 56.8) | 23.2 (17.2, 42.8) | 26.1 (20.0, 70.4) | 68.2 (49.0, 83.3) |
| m E | H | background | 26.8 (18.4, 36.9) | 25.6 (15.2, 41.3) | 8.8 (6.5, 21.6) | 13.7 (8.6, 35.3) |
| m E | H | viewpoint | 42.9 (34.2, 62.6) | 14.0 (10.5, 34.2) | 14.5 (10.4, 33.9) | 27.2 (22.8, 28.4) |
| m E | H | colour | 49.4 (36.3, 64.3) | 14.9 (4.8, 38.3) | 21.3 (9.5, 30.7) | 38.0 (26.8, 43.4) |
| m E | H | all | 18.6 (12.7, 22.6) | 14.4 (8.7, 19.5) | 11.2 (7.6, 18.6) | 34.0 (26.9, 41.5) |
| m M | E | background | 153.7 (100.4, 180.1) | 111.4 (47.2, 122.0) | 177.5 (95.8, 202.9) | 308.2 (266.4, 351.5) |
| m M | E | viewpoint | 41.1 (30.9, 65.8) | 42.5 (32.6, 59.7) | 34.6 (20.4, 66.7) | 84.1 (53.1, 97.6) |
| m M | E | colour | 115.7 (110.3, 148.7) | 94.3 (48.1, 132.3) | 101.7 (37.3, 145.1) | 244.4 (207.2, 290.8) |
| m M | E | all | 37.6 (31.0, 53.2) | 18.6 (11.9, 26.8) | 27.0 (17.1, 41.6) | 45.5 (36.0, 97.9) |
| m M | M | background | 49.5 (44.9, 91.6) | 20.8 (14.8, 63.5) | 34.6 (22.0, 54.0) | 61.1 (40.7, 76.5) |
| m M | M | viewpoint | 72.4 (57.3, 85.4) | 56.0 (47.4, 64.9) | 20.6 (18.8, 71.3) | 76.3 (43.5, 90.0) |
| m M | M | colour | 94.1 (65.7, 98.7) | 66.2 (53.0, 91.8) | 46.6 (35.9, 85.4) | 88.0 (56.8, 108.5) |
| m M | M | all | 40.3 (23.6, 52.5) | 14.3 (8.7, 17.7) | 15.6 (13.5, 19.8) | 50.3 (27.5, 52.9) |
| m M | H | background | 27.8 (21.0, 30.5) | 22.1 (12.3, 34.4) | 8.4 (6.7, 17.0) | 12.3 (7.3, 31.6) |
| m M | H | viewpoint | 29.2 (27.4, 60.3) | 13.3 (8.8, 36.4) | 25.4 (12.3, 34.6) | 23.4 (18.5, 24.9) |
| m M | H | colour | 51.2 (32.7, 58.8) | 12.4 (6.2, 40.4) | 18.5 (9.2, 31.8) | 35.4 (27.9, 43.9) |
| m M | H | all | 21.2 (14.4, 29.6) | 9.8 (7.2, 20.6) | 8.1 (7.2, 29.1) | 26.5 (19.7, 38.0) |
| m H | E | background | 121.6 (95.2, 163.5) | 76.5 (51.6, 108.0) | 105.4 (58.2, 143.6) | 274.8 (223.7, 310.8) |
| m H | E | viewpoint | 26.0 (23.6, 42.4) | 28.0 (15.1, 38.3) | 12.1 (6.3, 25.1) | 28.7 (17.9, 39.0) |
| m H | E | colour | 120.4 (109.5, 128.8) | 88.5 (57.3, 132.4) | 69.5 (35.2, 124.5) | 245.0 (162.3, 279.2) |
| m H | E | all | 26.3 (19.3, 33.6) | 21.8 (13.0, 31.3) | 8.7 (5.5, 14.5) | 18.5 (16.0, 24.8) |
| m H | M | background | 42.7 (37.5, 47.7) | 19.8 (11.0, 31.9) | 38.1 (23.7, 48.3) | 39.8 (25.4, 52.7) |
| m H | M | viewpoint | 41.6 (29.6, 60.9) | 32.2 (20.0, 42.3) | 12.5 (6.7, 18.8) | 38.3 (16.1, 64.9) |
| m H | M | colour | 85.5 (80.8, 90.0) | 67.3 (52.9, 76.2) | 46.8 (23.5, 49.9) | 75.6 (64.8, 94.5) |
| m H | M | all | 24.5 (20.4, 32.6) | 14.0 (7.1, 21.6) | 10.6 (7.6, 16.2) | 18.3 (14.4, 21.6) |
| m H | H | background | 25.5 (19.3, 28.5) | 20.9 (14.7, 27.3) | 8.9 (7.0, 17.8) | 16.8 (12.0, 20.4) |
| m H | H | viewpoint | 27.4 (25.0, 53.4) | 10.6 (6.9, 18.2) | 21.2 (11.4, 31.2) | 23.9 (20.2, 34.6) |
| m H | H | colour | 43.1 (26.7, 56.5) | 13.2 (6.8, 23.7) | 19.2 (7.7, 26.5) | 27.0 (24.2, 29.5) |
| m H | H | all | 33.7 (21.1, 37.6) | 12.8 (9.3, 16.2) | 15.6 (11.7, 21.2) | 16.2 (15.6, 19.9) |

# F   Ablation on Effectiveness of Dropblock

In this section, we investigate further the effect of dropblock on baseline algorithms. Firstly, the full results of ablation study on dropblock for our proposed algorithms is shown in Table 19. Secondly, an ablation study on using only dropblock with baseline algorithm DrQv2+BC is shown in Table 20. We can note that adding dropblock on baseline algorithm can improve the performance by a fair margin, but the full proposed algorithm consisting of dropblock and gradient reversal layer is the most performant.

Table 19: DropBlock ablation results on V-D4RL cheetah-run distraction dataset. The reported numbers are IQM and 95% stratified bootstrap CIs in between parentheses. Abbreviations for dataset type: medium: m, expert: e, medium-expert: m-e. Abbreviations for visual distractions difficulty: normal: N, easy: E, medium: M, hard: H. For example, "e H" equals "expert Hard".

**cheetah-run (V-D4RL)**

| train | eval | $p = 0.0$ | $p = 0.1$ | $p = 0.2$ | $p = 0.3$ | $p = 0.5$ |
|---|---|---|---|---|---|---|
| e E | N | 114.1 (93.2, 176.2) | 176.0 (149.2, 216.8) | 153.6 (61.5, 197.3) | 233.7 (197.9, 256.8) | 176.6 (82.2, 254.7) |
|  | E | 40.0 (26.5, 59.1) | 65.3 (52.9, 85.7) | 100.8 (83.6, 113.7) | 88.3 (72.1, 107.4) | 71.0 (64.0, 99.8) |
|  | M | 17.2 (16.0, 23.1) | 46.1 (30.0, 50.2) | 42.0 (35.6, 73.5) | 44.9 (34.3, 57.1) | 43.7 (35.5, 53.9) |
|  | H | 15.2 (11.2, 21.7) | 22.5 (18.1, 30.0) | 22.5 (20.4, 51.4) | 21.4 (17.3, 28.1) | 33.6 (22.3, 39.1) |
| e M | N | 81.8 (48.7, 105.8) | 46.9 (34.4, 78.4) | 29.9 (14.5, 70.2) | 76.5 (28.2, 110.3) | 33.9 (15.9, 60.7) |
|  | E | 49.1 (30.3, 58.7) | 30.2 (22.1, 35.5) | 31.1 (26.0, 51.1) | 32.8 (29.2, 37.4) | 30.8 (25.6, 50.9) |
|  | M | 30.2 (26.1, 47.4) | 27.7 (17.4, 38.5) | 22.6 (15.1, 30.6) | 23.0 (17.4, 27.4) | 27.7 (24.7, 41.5) |
|  | H | 21.6 (10.8, 23.2) | 11.2 (9.2, 27.5) | 20.1 (15.9, 27.3) | 20.1 (9.7, 24.0) | 17.7 (12.2, 20.5) |
| e H | N | 16.1 (8.0, 29.9) | 9.4 (5.9, 45.7) | 11.1 (6.6, 98.9) | 24.3 (17.8, 44.8) | 23.7 (17.7, 38.7) |
|  | E | 19.0 (15.6, 21.2) | 18.8 (12.6, 22.2) | 10.4 (7.5, 27.3) | 16.0 (12.8, 21.0) | 9.2 (6.5, 13.5) |
|  | M | 14.7 (11.9, 22.6) | 15.6 (9.9, 17.5) | 10.8 (7.5, 19.3) | 16.2 (12.9, 18.8) | 9.0 (6.4, 16.4) |
|  | H | 16.2 (13.6, 26.5) | 11.5 (9.0, 19.1) | 12.4 (9.8, 18.5) | 9.8 (7.5, 11.9) | 9.8 (6.1, 13.0) |
| m E | N | 134.1 (100.6, 174.2) | 181.3 (75.5, 249.6) | 101.3 (82.6, 141.6) | 246.9 (219.7, 279.7) | 104.4 (41.0, 230.8) |
|  | E | 82.0 (69.0, 100.3) | 127.2 (105.2, 153.1) | 123.1 (107.3, 182.2) | 152.0 (122.0, 178.9) | 114.4 (91.3, 136.9) |
|  | M | 66.0 (31.7, 73.6) | 68.2 (49.0, 83.3) | 58.2 (55.6, 63.4) | 64.0 (52.2, 77.8) | 44.1 (34.3, 52.7) |
|  | H | 28.8 (27.5, 39.3) | 32.8 (28.3, 40.1) | 28.0 (20.4, 32.4) | 30.7 (24.6, 35.7) | 34.0 (26.9, 41.5) |
| m M | N | 124.7 (59.7, 134.3) | 96.3 (82.4, 154.7) | 79.9 (46.2, 93.2) | 116.4 (68.0, 154.3) | 62.8 (23.3, 105.7) |
|  | E | 49.4 (31.2, 59.4) | 49.9 (31.4, 66.6) | 35.5 (28.4, 61.1) | 45.5 (36.0, 97.9) | 48.5 (38.5, 56.2) |
|  | M | 36.3 (27.1, 67.1) | 31.3 (19.9, 41.7) | 50.3 (27.5, 52.9) | 30.7 (21.1, 40.4) | 35.4 (31.2, 40.2) |
|  | H | 25.5 (20.7, 29.8) | 8.0 (6.6, 26.0) | 26.5 (19.7, 38.0) | 23.9 (15.4, 37.4) | 12.8 (8.5, 16.4) |
| m H | N | 35.7 (25.8, 81.3) | 8.6 (4.6, 26.9) | 11.9 (7.5, 34.2) | 36.2 (27.6, 47.7) | 19.3 (10.6, 36.0) |
|  | E | 24.5 (22.2, 31.8) | 18.5 (16.0, 24.8) | 11.7 (7.9, 13.1) | 14.8 (8.7, 21.3) | 7.9 (4.6, 20.0) |
|  | M | 22.0 (19.5, 28.8) | 18.3 (14.4, 21.6) | 12.7 (11.5, 15.3) | 12.1 (7.6, 14.2) | 6.9 (5.5, 22.6) |
|  | H | 19.1 (14.9, 23.4) | 16.2 (15.6, 19.9) | 13.3 (8.4, 19.9) | 11.4 (7.7, 20.4) | 9.6 (6.6, 15.5) |
| m-e E | N | 85.4 (45.4, 119.5) | 185.9 (172.9, 243.9) | 171.7 (94.4, 216.1) | 229.7 (160.6, 251.5) | 194.3 (75.8, 208.5) |
|  | E | 71.7 (56.9, 89.9) | 130.6 (102.0, 161.0) | 121.7 (99.9, 141.2) | 110.6 (83.2, 141.2) | 96.8 (91.9, 118.6) |
|  | M | 42.4 (31.9, 48.8) | 68.0 (56.7, 84.9) | 94.0 (57.6, 102.6) | 81.0 (66.3, 109.3) | 77.0 (65.5, 82.0) |
|  | H | 34.3 (32.0, 36.0) | 50.9 (31.3, 53.4) | 49.1 (48.0, 55.4) | 55.4 (42.2, 60.5) | 42.3 (36.7, 46.7) |
| m-e M | N | 63.7 (35.7, 116.4) | 33.5 (14.3, 115.4) | 43.0 (4.5, 108.4) | 84.4 (36.2, 116.0) | 60.2 (26.1, 76.5) |
|  | E | 33.4 (27.8, 47.4) | 27.3 (22.6, 38.2) | 40.5 (31.1, 55.0) | 60.5 (33.0, 72.8) | 53.0 (39.8, 90.6) |
|  | M | 45.0 (31.9, 59.6) | 29.7 (21.0, 32.2) | 31.2 (25.1, 44.0) | 43.9 (22.0, 59.5) | 49.2 (37.0, 66.9) |
|  | H | 25.3 (22.1, 27.5) | 13.2 (10.4, 31.7) | 20.2 (13.8, 30.8) | 13.4 (9.0, 25.7) | 15.3 (12.1, 39.5) |
| m-e H | N | 21.9 (15.7, 39.4) | 6.0 (2.4, 7.7) | 21.5 (17.1, 39.7) | 39.2 (16.9, 60.8) | 24.4 (11.3, 50.0) |
|  | E | 12.0 (10.1, 21.9) | 16.4 (13.0, 21.0) | 29.4 (21.0, 41.2) | 21.5 (14.7, 48.6) | 18.0 (15.6, 23.5) |
|  | M | 14.9 (13.0, 16.0) | 18.6 (11.0, 22.5) | 24.9 (17.7, 28.5) | 19.0 (11.2, 26.2) | 15.0 (11.5, 19.2) |
|  | H | 17.1 (11.5, 23.2) | 18.1 (13.2, 22.4) | 27.3 (16.7, 33.6) | 12.7 (10.9, 22.9) | 12.8 (9.5, 21.0) |
| total |  | 1550.4 | 1736.0 | 1694.2 | **2162.9** | 1645.1 |

Table 20: Results comparison on cheetah-run distraction dataset (V-D4RL). The reported numbers are IQM and 95% stratified bootstrap CIs in between parentheses. Abbreviations: normal: N, dis-easy: E, dis-medium: M, dis-hard: H. e: expert. m: medium. m-e: medium-expert. To save space, we note DrQv2+BC as DB, and DrQv2+BC with dropblock (drop probability = $p$) is noted as DB-$p$.

| | | cheetah-run (V-D4RL) | | | | |
|---|---|---|---|---|---|---|
| train | eval | DB | DB-0.1 | DB-0.2 | DB-0.3 | Ours |
| e E | N | 108.7 (42.0, 132.9) | 70.6 (26.2, 113.4) | 33.5 (19.9, 92.7) | 88.6 (48.3, 138.5) | **233.7** (197.9, 256.8) |
| | E | 34.2 (30.5, 47.3) | 42.2 (35.9, 75.1) | 52.9 (40.3, 65.5) | 60.1 (51.9, 74.5) | **100.8** (83.6, 113.7) |
| | M | 13.7 (11.6, 21.7) | 27.7 (18.0, 34.6) | 23.4 (15.3, 35.8) | 26.6 (19.3, 33.4) | **42.0** (35.6, 73.5) |
| | H | 9.7 (7.4, 12.7) | 14.7 (8.2, 18.6) | 12.9 (9.2, 16.8) | 19.3 (14.6, 28.5) | **33.6** (22.3, 39.1) |
| e M | N | 18.9 (4.2, 64.7) | 14.4 (4.9, 72.3) | 54.3 (20.0, 128.8) | 45.2 (20.9, 53.6) | **76.5** (28.2, 110.3) |
| | E | 33.3 (18.5, 40.7) | 24.1 (13.4, 35.2) | 14.4 (12.8, 23.4) | 22.6 (18.8, 35.7) | 31.1 (26.0, 51.1) |
| | M | 28.3 (22.7, 39.3) | 19.8 (14.4, 25.4) | 20.9 (18.8, 25.9) | 19.8 (14.9, 28.5) | 27.7 (24.7, 41.5) |
| | H | 11.1 (9.3, 14.8) | 9.9 (6.5, 15.6) | 9.5 (5.7, 11.3) | 11.1 (8.4, 13.7) | **20.1** (15.9, 27.3) |
| e H | N | 1.5 (1.0, 3.4) | 3.1 (2.1, 4.9) | 8.4 (7.1, 65.8) | 8.7 (5.7, 11.5) | **24.3** (17.8, 44.8) |
| | E | 9.1 (8.4, 14.2) | 9.6 (5.8, 9.8) | 8.6 (7.5, 13.0) | 9.9 (7.1, 11.6) | **18.8** (12.6, 22.2) |
| | M | 9.8 (7.9, 17.8) | 8.0 (6.2, 12.3) | 9.0 (6.5, 11.6) | 5.9 (5.0, 9.3) | **16.2** (12.9, 18.8) |
| | H | 11.1 (7.8, 17.2) | 7.0 (5.4, 8.6) | 8.0 (6.9, 10.4) | 6.4 (5.0, 10.4) | **12.4** (9.8, 18.5) |
| m E | N | 131.0 (115.0, 143.2) | 49.9 (17.0, 73.2) | 133.0 (58.6, 219.4) | **271.7** (110.1, 287.3) | 246.9 (219.7, 279.7) |
| | E | 99.5 (65.6, 110.4) | 108.6 (105.3, 122.9) | 111.9 (102.5, 150.0) | 115.8 (97.8, 134.2) | **152.0** (122.0, 178.9) |
| | M | 41.4 (32.4, 56.8) | 42.7 (37.7, 50.7) | 60.1 (48.2, 68.4) | 36.8 (26.9, 44.9) | **68.2** (49.0, 83.3) |
| | H | 18.6 (12.7, 22.6) | 19.7 (11.6, 25.1) | 22.2 (16.9, 27.3) | 14.9 (11.0, 40.8) | **34.0** (26.9, 41.5) |
| m M | N | **125.9** (81.9, 145.0) | 113.3 (35.3, 188.6) | 100.7 (59.1, 138.2) | 119.1 (41.1, 163.2) | 116.4 (68.0, 154.3) |
| | E | 37.6 (31.0, 53.2) | 34.6 (30.4, 38.8) | **54.3** (42.7, 74.1) | 45.5 (28.8, 59.8) | 45.5 (36.0, 97.9) |
| | M | 40.3 (23.6, 52.5) | 30.2 (24.8, 39.0) | 47.3 (19.7, 51.1) | 25.8 (22.6, 31.3) | **50.3** (27.5, 52.9) |
| | H | 21.2 (14.4, 29.6) | 21.0 (16.6, 24.5) | 19.7 (15.5, 27.1) | 9.9 (8.6, 45.1) | **26.5** (19.7, 38.0) |
| m H | N | **60.3** (43.5, 70.7) | 6.1 (4.6, 15.4) | 15.2 (11.7, 21.1) | 12.3 (8.2, 49.3) | 36.2 (27.6, 47.7) |
| | E | **26.3** (19.3, 33.6) | 15.5 (9.3, 18.3) | 11.4 (8.2, 15.0) | 4.7 (3.6, 7.9) | 18.5 (16.0, 24.8) |
| | M | **24.5** (20.4, 32.6) | 10.1 (8.2, 19.4) | 8.1 (6.1, 10.9) | 7.5 (5.1, 8.9) | 18.3 (14.4, 21.6) |
| | H | **33.7** (21.1, 37.6) | 14.5 (8.3, 21.6) | 13.3 (9.7, 18.9) | 4.5 (3.7, 5.2) | 16.2 (15.6, 19.9) |
| m-e E | N | 115.1 (51.4, 177.6) | 119.6 (43.4, 154.5) | 184.3 (85.2, 228.1) | 171.7 (86.8, 223.0) | **229.7** (160.6, 251.5) |
| | E | 83.6 (57.9, 96.0) | 114.4 (81.3, 134.0) | **141.4** (106.7, 152.5) | 93.6 (84.5, 117.7) | 130.6 (102.0, 161.0) |
| | M | 35.0 (33.1, 44.5) | 45.8 (37.8, 55.1) | 71.6 (56.3, 82.9) | 67.2 (55.9, 92.2) | **81.0** (66.3, 109.3) |
| | H | 18.1 (14.2, 28.0) | 30.7 (24.1, 50.1) | 31.8 (27.2, 43.2) | 38.9 (35.4, 47.7) | **55.4** (42.2, 60.5) |
| m-e M | N | 69.6 (48.1, 87.2) | 11.4 (6.6, 109.5) | 41.4 (8.9, 104.4) | **86.1** (32.6, 133.1) | 84.4 (36.2, 116.0) |
| | E | 32.5 (22.3, 56.2) | 27.6 (15.4, 38.2) | 40.8 (29.2, 62.9) | 35.8 (27.2, 40.7) | **60.5** (33.0, 72.8) |
| | M | 34.4 (21.3, 49.4) | 24.1 (17.9, 30.1) | 24.3 (19.3, 36.2) | 39.6 (24.0, 45.6) | **49.2** (37.0, 66.9) |
| | H | 13.5 (12.5, 18.6) | 15.3 (8.1, 22.1) | 12.4 (9.8, 25.1) | 9.6 (5.2, 13.7) | **20.2** (13.8, 30.8) |
| m-e H | N | 17.3 (9.9, 29.2) | 2.4 (1.7, 6.7) | 21.1 (11.0, 39.0) | 22.5 (10.4, 43.1) | **39.2** (16.9, 60.8) |
| | E | 14.2 (9.8, 16.5) | 9.8 (7.5, 13.7) | 17.6 (14.2, 44.9) | 12.3 (9.2, 18.6) | **29.4** (21.0, 41.2) |
| | M | 11.6 (10.4, 22.3) | 7.0 (6.5, 11.0) | 12.3 (9.7, 33.9) | 12.3 (9.1, 14.8) | **24.9** (17.7, 28.5) |
| | H | 12.4 (9.1, 15.7) | 7.5 (5.4, 10.3) | 12.1 (8.9, 15.2) | 10.0 (8.2, 18.7) | **27.3** (16.7, 33.6) |
| IQM total | | 1407.0 | 1132.9 | 1464.1 | 1592.3 | **2298.0** |

## G Robustness analysis

To show the robustness of the proposed method over baseline, we perform two additional analysis, all based on training with cheetah-run medium-expert easy dataset. Firstly, We calculated the Wasserstein distance between the encoded features of from different level of distractions. The Wasserstein distance can be seen as a similarity metric between the two encoded features. Specifically, we have two scenarios. One is sampling a batch of images from the V-D4RL cheetah-run medium-expert dataset, and another is collected via online environment (cheetah-run) by using a random policy, to collect online distractions not existing on the dataset. The values are averaged over the batch of observations (batch size = 256).

Table 21: Wasserstein distance calculated using different data sources. In the "distance between" column, all distance are compared between a distraction observation and a normal observation.

| cheetah-run medium-expert (V-D4RL) | | | |
|---|---|---|---|
| data source | distance between | DrQv2+BC | Ours |
| offline | normal - dis-easy | 0.5577 | **0.1413** |
| offline | normal - dis-medium | 0.5346 | **0.1709** |
| offline | normal - dis-hard | 0.2832 | **0.0887** |
| online | normal - dis-easy | 0.3732 | **0.1093** |
| online | normal - dis-medium | 0.3311 | **0.0483** |
| online | normal - dis-hard | 0.3312 | **0.1156** |

Secondly, we plot the t-SNE (van der Maaten, 2008) results of the trained policy distribution using different distraction observations. There are also two scenario of data source; from the dataset and one from online environment. As we can observe, in both scenarios, our proposed method brings the policy distribution closer for different distraction observations.

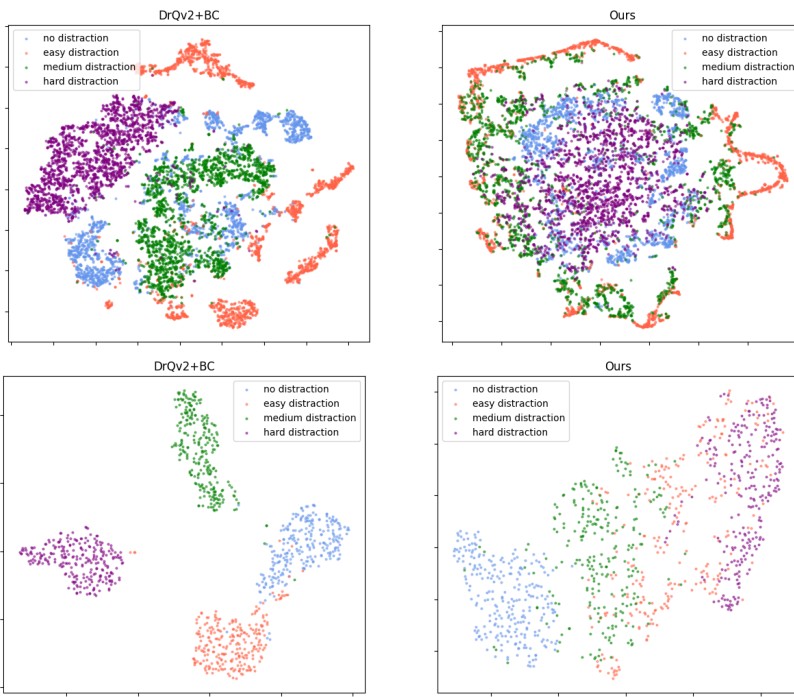

Figure 5: t-SNE plot on the policy distribution. Blue dots are normal observation. Red dots are easy distraction observations. Green dots are medium distraction. Purple dots are hard distraction.

# H    Additional Benchmark Comparison with V-D4RL

V-D4RL was collected by training an online SAC agent and using the saved checkpoints to collect visual observations while running the agent in proprioceptive states. As we could not access the SAC checkpoints of the original V-D4RL authors, we followed their guidelines as closely as possible to reproduce data collection. The checkpoint used to collect the medium dataset is described as the first stable checkpoint, resulting in consistent 500 rewards. Simultaneously, the expert agent is the stable checkpoint resulting in nearly the maximum possible reward, which translates to nearly 1000 rewards in DM Control tasks.

Figure 6 presents the histogram of collected cheetah-run medium-expert data, plotted by rewards per episode versus frequencies (counts). The figure shows that the original data and broad data are comparable in most parts; the main difference is that the medium difficulty of the broad dataset is a bit lower compared to the original, and the expert difficulty is significantly concentrated in the final rewards (around 900). The difference is because of the checkpoint used to gather data.

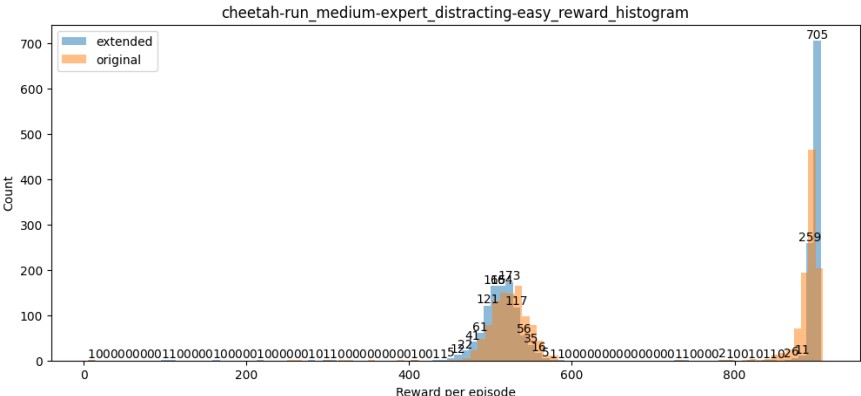

Figure 6: The original dataset is in orange, while the broad dataset is in blue.

Table 22: Full summary statistics of per-episode return in the V-D4RL vs our proposed dataset.

| Dataset | task | difficulty | Timesteps | Mean | Std. | Min | P25 | Median | P75 | Max |
|---|---|---|---|---|---|---|---|---|---|---|
| V-D4RL | cheetah-run | random | 100k | 6.9 | 2.6 | 1.7 | 4.8 | 6.4 | 8.6 | 15.9 |
| | | medexp | 200k | 707.5 | 186.0 | 253.8 | 527.7 | 710.6 | 894.1 | 905.7 |
| | | medium | 100k | 523.3 | 32.9 | 253.8 | 511.1 | 527.6 | 539.2 | 578.3 |
| | | expert | 100k | 891.6 | 11.1 | 843.0 | 888.3 | 894.1 | 899.2 | 905.7 |
| | walker-walk | random | 100k | 40.1 | 7.9 | 28.1 | 33.8 | 38.6 | 44.2 | 66.3 |
| | | medexp | 200k | 703.7 | 270.4 | 65.6 | 444.4 | 739.6 | 969.6 | 990.6 |
| | | medium | 100k | 436.8 | 57.4 | 65.6 | 419.2 | 444.2 | 469.0 | 538.6 |
| | | expert | 100k | 970.6 | 11.1 | 940.6 | 964.2 | 969.7 | 979.2 | 990.6 |
| Ours | cheetah-run | random | 100k | 6.9 | 2.6 | 1.7 | 4.8 | 6.4 | 8.6 | 15.9 |
| | | medexp | 200k | 705.9 | 192.8 | 447.3 | 514.6 | 695.6 | 899.4 | 905.1 |
| | | medium | 100k | 514.0 | 21.3 | 447.3 | 501.3 | 514.4 | 525.9 | 582.1 |
| | | expert | 100k | 897.8 | 8.7 | 809.0 | 896.9 | 899.3 | 901.2 | 905.1 |
| | walker-walk | random | 100k | 40.0 | 7.9 | 28.3 | 33.8 | 38.4 | 44.4 | 66.1 |
| | | medexp | 200k | 723.6 | 198.0 | 39.1 | 547.9 | 710.1 | 915.3 | 972.6 |
| | | medium | 100k | 535.7 | 81.2 | 39.1 | 522.3 | 547.8 | 572.7 | 661.1 |
| | | expert | 100k | 911.5 | 32.1 | 759.0 | 896.4 | 915.3 | 935.2 | 972.6 |
| | reacher-easy | random | 100k | 52.2 | 81.9 | 0.0 | 0.0 | 4.0 | 81.0 | 378.0 |
| | | medexp | 200k | 738.5 | 385.2 | 0.0 | 514.8 | 961.0 | 979.0 | 1000.0 |
| | | medium | 100k | 502.5 | 430.6 | 0.0 | 10.2 | 512.5 | 952.0 | 1000.0 |
| | | expert | 100k | 974.5 | 13.7 | 920.0 | 964.0 | 975.0 | 985.0 | 1000.0 |
| | reacher-hard | random | 100k | 10.4 | 22.2 | 0.0 | 0.0 | 0.0 | 12.0 | 167.0 |
| | | medexp | 200k | 845.1 | 296.1 | 0.0 | 946.0 | 961.0 | 976.0 | 999.0 |
| | | medium | 100k | 735.0 | 374.5 | 0.0 | 583.2 | 949.0 | 967.2 | 999.0 |
| | | expert | 100k | 955.2 | 105.7 | 4.0 | 958.0 | 969.5 | 980.0 | 999.0 |
| | cup-catch | random | 100k | 52.3 | 190.7 | 0.0 | 0.0 | 0.0 | 0.0 | 846.0 |
| | | medexp | 200k | 931.1 | 91.0 | 431.0 | 937.0 | 960.5 | 986.0 | 1000.0 |
| | | medium | 100k | 891.3 | 114.4 | 431.0 | 859.8 | 937.5 | 962.0 | 1000.0 |
| | | expert | 100k | 970.9 | 18.4 | 937.0 | 953.0 | 964.0 | 990.0 | 1000.0 |

# I Implementation Details

The implementation details and hyperparameters settings used for our experiments are presented below. Similar to DrQv2, we employed clipped double Q-learning (Fujimoto et al., 2018) to reduce the overestimation bias in the target value. We also provide a schematic of how the overall network architecture is implemented as in Figure 3, as well as the network architecture used in Table 28. The Actor and critic networks follow DrQv2 exactly; therefore, they are omitted. Additionally, we show the hyperparameters used in the experiments.

Table 23 lists the common hyperparameters for all methods. Table 24 lists the additional hyperparameters used in our proposed method. Table 25 lists the additional hyperparameters used in the experiments for the baseline DrQv2+BC. Table 26 lists the additional hyperparameters used in the experiments for the baseline AWAC+BC. Table 27 lists the additional hyperparameters used in the experiments for the baseline IQL.

For PAD (Hansen et al., 2021b), ILA (Yoneda et al., 2022) and SVEA Hansen et al. (2021a), we use their default hyperparameters. We include some brief explanation on some of the baseline algorithms to complement the hyperparameters listed in Appendix I.4.

## I.1 DrQv2+BC

DrQv2+BC (Lu et al., 2023) is a modified version of DrQv2 (Yarats et al., 2022) by adding a BC term into the policy network training. The term is regulated by a hyperparameter $\lambda_{\mathrm{bc}}$. The usage of $\lambda_{\mathrm{bc}}$ is as:

$$\pi = \underset{\pi}{\mathrm{argmax}}\, \mathbb{E}_{(s_t,a_t)\sim\mathcal{D},a^\pi\sim\pi(\cdot|f(s;\theta_f))} \left[ Q(f(s_t;\theta_f),a_t^\pi;\,\theta_q) - \lambda_{\mathrm{bc}}(a_t^\pi - a_t)^2 \right]$$

## I.2 AWAC

AWAC updates its policy by a weighted maximum likelihood, where the targets are obtained by reweighting the state-action pairs observed in the dataset versus the predicted advantages from the critic, $(Q(f(s_t;\theta_f),a_t) - Q(f(s_t;\theta_f),a_t^\pi);\,\theta_q))$, where $a^\pi \sim \pi(\cdot|f(s;\theta_f))$. By adding a BC term into the policy training, the policy objective is calculated as:

$$\pi = \underset{\pi}{\mathrm{argmax}}\, \mathbb{E}_{(s_t,a_t)\sim\mathcal{D},a_t^\pi\sim\pi(\cdot|f(s;\theta_f))}$$

$$\left[ \left[ \log a^\pi \exp(\frac{1}{\lambda_{\mathrm{awac}}}(Q(f(s_t;\theta_f),a_t;\theta_q) - Q(f(s_t;\theta_f),a_t^\pi;\theta_q))) \right] - \lambda_{\mathrm{bc}}(a_t^\pi - a_t)^2 \right]$$

## I.3 IQL

IQL uses the same advantage-weighted regression policy training as AWAC while renaming the hyperparameter to a more straightforward scale $\beta$. In our experiments, we did not include an additional BC term for IQL. The policy objective for IQL is:

$$\pi = \underset{\pi}{\mathrm{argmax}}\, \mathbb{E}_{(s_t,a_t)\sim\mathcal{D},a_t^\pi\sim\pi(\cdot|f(s;\theta_f))}[[\log a^\pi \exp(\beta(\,Q(f(s_t;\theta_f),a_t;\theta_q) - V(f(s_t;\theta_f);\theta_v)))]$$

For the expectile regression loss, IQL minimises the following two losses, where $\tau_{\mathrm{IQL}}$ is given as a hyperparameter for expectile $\ell_{\mathrm{IQL}}^2$:

$$\min_{Q} L_{\mathrm{IQL}}^{Q} = \mathbb{E}_{(s_t,a_t,s_{t+1})\sim\mathcal{D}} \left[ (r + \gamma V(f(s_{t+1};\theta_f);\theta_v) - Q(f(s_t;\theta_f),a_t))^2 \right]$$

$$\min_{V} L_{\mathrm{IQL}}^{V} = \mathbb{E}_{(s_t,a_t)\sim\mathcal{D}} \left[ \ell_{\mathrm{IQL}}^2(\overline{Q}(f(s_t;\theta_f),a_t) - V(f(s_t;\theta_f))) \right], \quad \text{where } \ell_{\mathrm{IQL}}^2 = |\tau_{\mathrm{IQL}} - \mathbb{1}(x < 0)|x^2$$

### I.4 Hyperparameters

Table 23: Common hyperparameters.

| Parameters | Value | | Parameters | Value |
|---|---|---|---|---|
| optimiser (all networks) | Adam | | batch size | 256 |
| encoder learning rate | $3e-04$ | | critic learning rate | $3e-04$ |
| actor learning rate | $3e-04$ | | feature dim | 50 |
| activation function | ReLU | | hidden dim | 1024 |
| frame stacking (all tasks) | 3 | | action repeat (all tasks) | 2 |
| nstep (all tasks) | 3 | | weight decay | 0 |
| critic target tau ($\tau$) | 0.01 | | | |

Table 24: Additional hyperparameters for our proposed algorithm.

| Parameters | Value | | Parameters | Value |
|---|---|---|---|---|
| behaviour cloning (BC) weight $\lambda_{\text{bc}}$ | 2.5 | | gradient reversal $\mu$ | -1 |
| dropblock drop rate | Table 29 | | discriminator learning rate | $1e-05$ |

Table 25: Additional hyperparameters for baseline DrQv2+BC.

| Parameters | Value |
|---|---|
| behaviour cloning (BC) weight $\lambda_{\text{bc}}$ | 2.5 |

Table 26: Additional hyperparameters for baseline AWAC+BC.

| Parameters | Value |
|---|---|
| AWAC lambda $\lambda_{\text{awac}}$ | 0.3 |

Table 27: Additional hyperparameters for baseline IQL.

| Parameters | Value | | Parameters | Value |
|---|---|---|---|---|
| IQL scale $\beta$ | 3 | | IQL expectile $\tau_{\text{IQL}}$ | 0.7 |

### I.5 Revelant Network Structures

Table 28: Network architectures

| Network name | Operations | Kernel | Strides | Ch I/O |
|---|---|---|---|---|
| | Conv + ReLU | 3x3 | 2 | 9/32 |
| | DropBlock | - | - | - |
| | Conv + ReLU | 3x3 | 1 | 32/32 |
| Encoder | DropBlock | - | - | - |
| | Conv + ReLU | 3x3 | 1 | 32/32 |
| | DropBlock | - | - | - |
| | Conv + ReLU | 3x3 | 1 | 32/32 |
| | Linear + ReLU | - | - | 39200/1024 |
| Discriminator | Linear + ReLU | - | - | 1024/50 |
| | Linear + ReLU | - | - | 50/1 |

### I.6 DropBlock hyperparameters

Table 29: DropBlock rate $p$ for all layers in the encoder for all tasks of the results in Table 8, Table 10, Table 11, Table 12, Table 13 and Table 9. Abbreviations for dataset type: medium: m, expert: e, medium-expert: m-e. Abbreviations for visual distractions difficulty: normal: N, easy: E, medium: M, hard: H.

| Task | eval | cheetah-run | cheetah-run (add.) | ball-catch | reacher-easy | reacher-hard | walker-walk |
|------|------|-------------|--------------------|------------|--------------|--------------|-------------|
| e E | N | 0.3 | 0.5 | 0.5 | 0.5 | 0.5 | 0.2 |
| e E | E | 0.2 | 0.3 | 0.5 | 0.5 | 0.5 | 0.1 |
| e E | M | 0.2 | 0.3 | 0.5 | 0.5 | 0.5 | 0.2 |
| e E | H | 0.5 | 0.3 | 0.5 | 0.5 | 0.5 | 0.2 |
| e M | N | 0.3 | 0.1 | 0.3 | 0.3 | 0.5 | 0.1 |
| e M | E | 0.2 | 0.1 | 0.3 | 0.5 | 0.5 | 0.2 |
| e M | M | 0.5 | 0.1 | 0.5 | 0.5 | 0.5 | 0.3 |
| e M | H | 0.2 | 0.1 | 0.5 | 0.5 | 0.5 | 0.1 |
| e H | N | 0.3 | 0.5 | 0.5 | 0.3 | 0.2 | 0.1 |
| e H | E | 0.1 | 0.5 | 0.5 | 0.5 | 0.2 | 0.1 |
| e H | M | 0.3 | 0.5 | 0.5 | 0.3 | 0.2 | 0.1 |
| e H | H | 0.2 | 0.2 | 0.5 | 0.3 | 0.2 | 0.1 |
| m E | N | 0.3 | 0.1 | 0.2 | 0.1 | 0.2 | 0.2 |
| m E | E | 0.3 | 0.1 | 0.3 | 0.3 | 0.1 | 0.2 |
| m E | M | 0.1 | 0.5 | 0.5 | 0.2 | 0.5 | 0.2 |
| m E | H | 0.5 | 0.5 | 0.3 | 0.2 | 0.5 | 0.3 |
| m M | N | 0.1 | 0.1 | 0.2 | 0.3 | 0.1 | 0.5 |
| m M | E | 0.3 | 0.3 | 0.1 | 0.3 | 0.3 | 0.5 |
| m M | M | 0.2 | 0.5 | 0.2 | 0.3 | 0.2 | 0.5 |
| m M | H | 0.2 | 0.5 | 0.1 | 0.1 | 0.2 | 0.2 |
| m H | N | 0.3 | 0.5 | 0.5 | 0.5 | 0.2 | 0.1 |
| m H | E | 0.1 | 0.5 | 0.1 | 0.3 | 0.2 | 0.1 |
| m H | M | 0.1 | 0.5 | 0.1 | 0.5 | 0.1 | 0.2 |
| m H | H | 0.1 | 0.2 | 0.2 | 0.2 | 0.2 | 0.1 |
| m-e E | N | 0.3 | 0.5 | 0.3 | 0.1 | 0.3 | 0.5 |
| m-e E | E | 0.1 | 0.5 | 0.2 | 0.3 | 0.5 | 0.2 |
| m-e E | M | 0.3 | 0.1 | 0.3 | 0.3 | 0.5 | 0.3 |
| m-e E | H | 0.3 | 0.2 | 0.2 | 0.2 | 0.1 | 0.5 |
| m-e M | N | 0.3 | 0.5 | 0.1 | 0.1 | 0.3 | 0.1 |
| m-e M | E | 0.3 | 0.5 | 0.2 | 0.5 | 0.5 | 0.1 |
| m-e M | M | 0.5 | 0.3 | 0.2 | 0.1 | 0.5 | 0.2 |
| m-e M | H | 0.2 | 0.5 | 0.3 | 0.1 | 0.5 | 0.2 |
| m-e H | N | 0.3 | 0.5 | 0.5 | 0.2 | 0.1 | 0.1 |
| m-e H | E | 0.2 | 0.3 | 0.1 | 0.2 | 0.5 | 0.2 |
| m-e H | M | 0.2 | 0.3 | 0.5 | 0.3 | 0.1 | 0.1 |
| m-e H | H | 0.2 | 0.3 | 0.1 | 0.1 | 0.3 | 0.3 |

