# OpenReview forum: "Offline Deep Reinforcement Learning for Visual Distractions via Domain Adversarial Training"
_TMLR — Accepted by TMLR_

### Review · Reviewer_B56R · 2024-07-29

**Summary Of Contributions:**

This paper proposes an adversarial approach for dealing with visual distractors in offline RL from images (and an architectural change to add drop-block to the encoder). Using an expanded version of the V-D4RL benchmark, the method is compared favorably with prior work of model-free offline RL from images (e.g. using image augmentations as in DrQv2+BC) and some methods for domain adaptation from online RL.

**Audience:**

Yes

**Claims And Evidence:**

No

**Requested Changes:**

1. Add a model based baseline as in the V-D4RL paper.

2. Clarify the data used for each of the baselines.

3. Add more ablations of dropblock on top of the baselines.

**Strengths And Weaknesses:**

### Strengths

1. The paper is well-written and clear. The motivation for the proposed method is well explained and the method itself seems to make sense.

2. The reported experimental results are in general strong. It does seem that the full method is outperforming the baselines. There are some reasonable ablations presented to rationalize design decisions.

### Weaknesses

1. The original V-D4RL paper finds more robust performance to distractors by using a model-based method (their version of offline Dreamer V2). However, this paper does not compare to this method, or any model based method. It would make the paper more comprehensive to compare to this sort of baseline.

2. To my understanding, the baseline methods are trained only on the distraction dataset while the visual encoder for the proposed method is trained using the distraction and the normal dataset. This seems to be an asymmetry that is not very well controlled for in the experimental methodology. Perhaps I misunderstand something, but more clarity is needed here and perhaps another ablation where the baselines all have access to the same data is warranted.

3. The use of drop-block somewhat confounds the results of the adversarial loss. There is an ablation in table 6, but there it does seem that if we remove drop-block, then the proposed method may not actually be much different from the baselines in table 2. A more direct comparison here would be good. Or even better, since drop-block is just an architectural change, it could easily be applied to the standard baselines (e.g. DrQv2+BC) to see if just adding drop-block alone makes these algorithms competitive with the proposed approach.

---

> ### Author Response · Authors · 2024-09-16
> **Authors' responses**
>
> We thank the reviewer for the constructive feedbacks. We are encouraged to find that the reviewer find our method intuitive and appreciate the questions regarding experimental setups. Responding to your questions:
>
>
> > 1. add offline model-based baseline
>
> Similarly to V-D4RL, we have added experiments on cheetah-run. The results are marked on red, and added to Table 2 and 3. We find it surprising that DreamerV2 (DV2) is so performant for cheetah-run, while is much less performant for walker-walk. In the original V-D4RL experiments, where the input is only normal observations, DV2 lags behind DrQv2+BC for their experiments. These are some interesting finding and warrant further investigation.
>
>
> > 2. regarding different data usages
>
> The experiments were designed to show that the encoder were indeed the bottleneck for performance, not the actor-critic part of the algorithm. Thus, as mentioned in the manuscript, for baseline methods and the proposed method, their actor-critic part only uses distracting data as training. Nevertheless, we do agree with the reviewer that there can be alternative experiment setups.
>
> To this end, we added additional experiments on where the baseline methods’ encoder is exposed to both distraction and non-distraction dataset. We have added an section in Appendix D (Table 15) to discuss this question.
> We find it quite interesting that when evaluating on normal observations, for expert dataset and medium dataset, the performance gain is significant, but it isn’t the case for medium-expert dataset as shown below.
>
> |train | eval  | drqv2_default | drqv2_tog | drqv2_both |
> | --- | --- | --- | --- | --- |
> m-e E  |normal | 115.1 (51.4, 177.6)  | 93.3 (35.5, 142.8) | 229.7 (160.6, 251.5) |
> m-e E  | easy | 83.6 (57.9, 96.0) | 96.8 (91.9, 118.6) | 69.6 (53.8, 129.7) |
> m-e E  | medium | 35.0 (33.1, 44.5) | 77.0 (65.5, 82.0) | 28.2 (20.8, 38.8) |
> m-e E | hard | 18.1 (14.2, 28.0) | 42.3 (36.7, 46.7) | 20.6 (16.0, 29.5) |
> m-e M  | normal | 69.6 (48.1, 87.2) | 60.2 (26.1, 76.5) | 88.2 (35.6, 137.2) |
> m-e M | easy | 32.5 (22.3, 56.2) | 53.0 (39.8, 90.6) | 33.1 (27.0, 54.0) |
> m-e M | medium | 34.4 (21.3, 49.4) | 49.2 (37.0, 66.9) | 23.5 (20.7, 26.1) |
> m-e M | hard | 13.5 (12.5, 18.6) | 15.3 (12.1, 39.5) | 13.5 (11.9, 29.3) |
> m-e H | normal | 17.3 (9.9, 29.2) | 24.4 (11.3, 50.0) | 7.7 (3.0, 18.2) |
> m-e H | easy | 14.2 (9.8, 16.5) | 18.0 (15.6, 23.5) | 14.5 (11.3, 19.2) |
> m-e H | medium | 11.6 (10.4, 22.3) | 15.0 (11.5, 19.2) | 16.3 (14.4, 20.6) |
> m-e H | hard | 12.4 (9.1, 15.7) | 12.8 (9.5, 21.0) | 15.1 (10.8, 19.4) |
>
>
> > 3. add ablation of dropblock
>
> In Appendix F Table 19, We have added an additional ablation study using cheetah-run (V-D4RL) dataset on the effectiveness of dropblock, varying the drop rate from 0.1 to 0.3, added to baseline DrQv2+BC. We can observe that adding dropblock to baseline DrQv2+BC does help a bit, overall having both the gradient reversal layer and dropblock show addictive performance gains. The medium-expert results are shown here below.
>
>
> |train | eval | drqv2_default | drqv2_drop0.1 | drqv2_drop0.2 | drqv2_drop0.3 | drqv2_tog |
> | --- | --- | --- | --- | --- | --- | --- |
> m-e E | normal | 115.1 (51.4, 177.6) | 119.6 (43.4, 154.5) | 184.3 (85.2, 228.1) | 171.7 (86.8, 223.0) | 194.3 (75.8, 208.5) |
> m-e E | easy | 83.6 (57.9, 96.0) | 114.4 (81.3, 134.0) | 141.4 (106.7, 152.5) | 93.6 (84.5, 117.7) | 96.8 (91.9, 118.6) |
> m-e E | medium | 35.0 (33.1, 44.5) | 45.8 (37.8, 55.1) | 71.6 (56.3, 82.9) | 67.2 (55.9, 92.2) | 77.0 (65.5, 82.0) |
> m-e E | hard | 18.1 (14.2, 28.0) | 30.7 (24.1, 50.1) | 31.8 (27.2, 43.2) | 38.9 (35.4, 47.7) | 42.3 (36.7, 46.7) |
> m-e M  | normal | 69.6 (48.1, 87.2) | 11.4 (6.6, 109.5) | 41.4 (8.9, 104.4) | 86.1 (32.6, 133.1) | 60.2 (26.1, 76.5) |
> m-e M | easy | 32.5 (22.3, 56.2) | 27.6 (15.4, 38.2) | 40.8 (29.2, 62.9) | 35.8 (27.2, 40.7) | 53.0 (39.8, 90.6) |
> m-e M | medium | 34.4 (21.3, 49.4) | 24.1 (17.9, 30.1) | 24.3 (19.3, 36.2) | 39.6 (24.0, 45.6) | 49.2 (37.0, 66.9) |
> m-e M | hard | 13.5 (12.5, 18.6) | 15.3 (8.1, 22.1) | 12.4 (9.8, 25.1) | 9.6 (5.2, 13.7) | 15.3 (12.1, 39.5) |
> m-e H | normal | 17.3 (9.9, 29.2) | 2.4 (1.7, 6.7) | 21.1 (11.0, 39.0) | 22.5 (10.4, 43.1) | 24.4 (11.3, 50.0) |
> m-e H | easy | 14.2 (9.8, 16.5) | 9.8 (7.5, 13.7) | 17.6 (14.2, 44.9) | 12.3 (9.2, 18.6) | 18.0 (15.6, 23.5) |
> m-e H | medium | 11.6 (10.4, 22.3) | 7.0 (6.5, 11.0) | 12.3 (9.7, 33.9) | 12.3 (9.1, 14.8) | 15.0 (11.5, 19.2) |
> m-e H | hard | 12.4 (9.1, 15.7) | 7.5 (5.4, 10.3) | 12.1 (8.9, 15.2) | 10.0 (8.2, 18.7) | 12.8 (9.5, 21.0) |

---

### Review · Reviewer_2QTB · 2024-08-01

**Summary Of Contributions:**

The paper "Offline Deep Reinforcement Learning for Visual Distractions via Domain Adversarial Training" presents a novel approach to enhance the robustness of reinforcement learning (RL) agents to visual distractions in offline settings. By introducing an adversarial training framework, the authors aim to train visual encoders to focus on domain-invariant features, thereby reducing the impact of visual distractions. The method involves using a domain discriminator to classify latent features into normal and distracted domains, and training the visual encoder to minimize the discrepancy between these features. Additionally, the paper extends the V-D4RL dataset to include more tasks with visual distractions and demonstrates the effectiveness of the proposed approach through extensive empirical evaluations, showing improvements over state-of-the-art baselines.

**Audience:**

Yes

**Broader Impact Concerns:**

No ethical concerns.

**Claims And Evidence:**

Yes

**Requested Changes:**

Include a comparison of the proposed method with third-person imitation learning approaches.

Evaluate the proposed method on a wider range of RL tasks beyond locomotion, such as manipulation or navigation tasks. This would strengthen the claims of the method's general robustness and applicability across different RL scenarios.

Provide a more detailed analysis of how the domain discriminator impacts feature extraction and the resulting policy robustness. Discuss the instability of this adversarial training, does techniques from generative AI domain such as those training GAN and stabilize GAN training approaches help for this work?

**Strengths And Weaknesses:**

Strengths
Novel Adversarial Training Framework: The paper introduces an innovative use of adversarial training to tackle visual distractions in offline reinforcement learning (RL). This approach focuses on learning domain-invariant features, which is a great advancement in making RL agents more robust to environmental variations.

Comprehensive Dataset Extension: The extension of the V-D4RL dataset to include more locomotion tasks with diverse visual distractions is a valuable contribution. This enhanced dataset provides a more robust benchmarking tool for evaluating visual-based RL methods.

Empirical Validation: The empirical results demonstrate that the proposed method consistently outperforms state-of-the-art baselines on tasks involving visual distractions. This strong performance across different datasets highlights the effectiveness of the approach.

Integration of DropBlock: The use of DropBlock in the visual encoder to prevent overfitting to specific features is a practical enhancement. This technique forces the encoder to rely on a broader set of features, improving the general robustness of the model.

Weaknesses

Evaluation Scope: While the paper extends the V-D4RL dataset, the evaluation is primarily focused on locomotion tasks. Evaluating the method on a broader range of RL tasks would strengthen the claims regarding its general robustness.

Lack of Comparison with Third-Person Imitation Learning: The paper does not compare the proposed method with third-person imitation learning approaches, nor does it mention relevant papers in this area. As these methods share very similar spirit, therefore, including such comparisons would provide a more comprehensive evaluation of the proposed method's effectiveness.

Lack of real world data: though this approach uses mostly synthetic environments, it remains unclear whether this approach would apply for real world problems where the visual distraction may appear in more complicated format.

---

> ### Author Response · Authors · 2024-09-16
> **Authors' responses**
>
> We want to thank the reviewer for the constructive feedback. We agree with the reviewer on we can extend the dataset to include tasks other then locomotion tasks and include into real-world data in the future. Regarding your questions:
>
> > Third-Person Imitation Learning
>
> We are not sure if the reviewer meant the paper of Third-Person Imitation Learning [1], thus we are basing our response on the paper. The approach does look similar, but we think that our problem settings is quite different as we are not particularly doing imitation learning. We think our work is more on the side of offline image-based RL, but we definitely hope we can further investigate this kind of problem settings in the future.
>
> > add navigation or manipulation tasks
>
> We thank the reviewer for the feedback on dataset construction. While we agree with the reviewer that to make the dataset even more complete manipulation or navigation tasks should also be included, we would like to point out that the original V-D4RL dataset includes only locomotion tasks.
>
> As we base all our experiments on the DM control suite, we tried to train SAC on the “Manipulator” tasks in DM control suite. Training a successful policy for the Manipulator tasks (manipulator-insert-peg, manipulator-insert-ball) is particularly difficult. Mentioned in RL unplugged [2], the only successful work that can solve manipulator tasks is V-MPO [3] (even D4PG [4] fail to learn the tasks). However, open source code are not provided by V-MPO authors, thus we would like to leave these more difficult tasks as future works.
>
> > Robustness analysis
>
> We are not very familiar with the GAN literature on how domain discriminator impacts feature extraction, so we attempt to answer the question from the point-of-view of 1) Wasserstein distance between encoder features and 2) brief analyses of the trained policy outputs using T-SNE. The results can be found in the updated manuscript Appendix G.
>
>
> [1] Bradly C. Stadie et al., Third-Person Imitation Learning, Arxiv 2017.
> [2] Caglar Gulcehre et al., RL Unplugged: A Suite of Benchmarks for Offline Reinforcement Learning, NeurIPS 2021.
> [3] H. Francis Song et al., V-MPO: On-Policy Maximum a Posteriori Policy Optimization for Discrete and Continuous Control, ICLR 2020
> [4] Gabriel Barth-Maron et al., Distributed Distributional Deterministic Policy Gradients, ICLR 2018

---

### Review · Reviewer_aSdA · 2024-09-03

**Summary Of Contributions:**

This paper follows up from (Lu et al, 2023) to study visual distractions in offline reinforcement learning using v-D4RL. Their contributions include
- Substantially extending the dataset from (Lu et al, 2023)
- Proposing a method to learn better policies in the face of visual distractions. Their method employs an adversarial approach which learns a discriminator on whether the environment contains distractions or not. The visual encoder is then also used to learn RL policies from offline data.
- Several experiments comparing with DrQv2+BC, IQL, and AWAC+BC. The paper also includes a parameter study of their new hyperparameter.

**Audience:**

Yes

**Claims And Evidence:**

Yes

**Requested Changes:**

I would recommend weaknesses 1-3 be addressed, while weakness 4 & 5 are something to think about if the authors have compute available.

**Strengths And Weaknesses:**

# Strengths

- The paper is mostly clear (see questions below), and the ideas presented are well motivated.
- The new dataset extensively expands the previous visual D4RL dataset in several domains with several different policies.
- The method is reasonable, with some reasonable flaws (see below).

# Weaknesses

I am fairly happy with the paper as presented, but have a few recommendations and questions.

- **W-1**. I don't think you ever disclose what the discriminator is discriminating in the dataset.  I believe it is whether the data has visual distractions or not, but it is unclear given the text in section 4.1. Just clarifying what the classes are would be sufficient.
- **W-2**. A major weakness of this method is the need for non-distracting data. It is very likely this will be difficult to get in any real-world domain. While I don't expect the authors to provide a solution in this paper, a section going over the weaknesses/limitations of the method might be warranted.
- **W-3**. You should disclose how the hyperparameters were tuned. While they are listed in detail (which I'm grateful for), I don't think the paper discloses how they were tuned. I think this should be in the main paper, and the main text should also point to the appendix with the hyperparameters to make sure readers know it exists or to go looking for it. If hyperparameters are re-used from old work, I believe this is another major weakness in the results shown. I highly recommend [this paper](https://arxiv.org/abs/2304.01315) for guiding your future empirical study design. This is a large weakness in the field in general.
- **W-4**. I think a missed opportunity in the ablation study was to ask whether the reverse gradient idea is additive in the RL tasks tested here. While I think it still would be (the reasoning is sound) and (Ganin and Lempitsky, 2015) paper does this in the supervised setting, I think it is low hanging fruit to see how much of a difference it makes here as well. I don't think it is necessary to include on all the domains and maybe just a single domain to show it is additive.
- **W-5**. I would have preferred to see [InAC](https://openreview.net/forum?id=u-RuvyDYqCM) tested as it has shown to be a strong baseline in several domains. I understand it might be difficult as the method hasn't been tested thoroughly in image domains so hyperparameters likely need to be tuned.

---

> ### Author Response · Authors · 2024-09-16
> **Authors' responses**
>
> We want to thank the reviewer for the constructive feedback. We are encouraged that the reviewer find the our paper clear and well motivated. Regarding your questions:
>
> > clarifying what the classes are
>
> The classes are binary labels, representing the normal observation (without distraction) and the distraction observation. We have cleared up the manuscript regarding this in Section 4.1.
>
> > add limitations
>
> We have added some discussion about the limitations of our work in Section 9. Another reviewer mentioned that current experiment comparison can be unfair as the encoder of the baseline algorithms uses only distraction data while the encoder of our proposed method use both normal and distraction data. Therefore, we have performed an additional study on the data usage by given both data to encoder of baseline algorithms.
>
> > regarding hyperparameters
>
> We thank the reviewer for pointing out the paper regarding empirical design in RL. In general, we have followed the hyperparameters of previous works of DrQv2, AWAC, IQL and V-D4RL experiments, but we do agree that this can be seen as a weakness in the field in general, especially given that the experiment settings are different from the original works.  We have added to the manuscript describing how these hyperparameters in Section 6.
>
> > addictiveness of gradient reversal layer
>
> We have experimented on this indirectly via ablation on dropblock probability (Table 6 and Table 19) where dropblock probability $p=0.0$ would result in only the gradient reversal layer. The original baseline is shown in Table 2 and Table 8. By comparing the tables, we can observe some performance gain by only adding the gradient reversal layer.
>
> > Experiments on InAC [1]
>
> We thank the reviewer for point us to the paper. We have tried to adapt the method, but somehow the training process is quite unstable using the current hyperparameters. We will continue to investigate into this.
>
> [1] Chenjun Xiao et al., The In-Sample Softmax for Offline Reinforcement Learning, ICLR 2023

---

### Author Response · Authors · 2024-09-16
**Update to manuscript**

Dear reviewers,

We are thankful for the reviewers' time into reviewing our paper. We are encouraged by the reviewers generally finding our work clear and well motivated. Regarding the questions, we have updated the manuscript and marked the additions in red in the updated version. Generally speaking, we have added a few ablation studies on the usage of dataset and a model-based baseline, as well as brief analysis on the robustness of learnt features from the encoder. We would greatly appreciate if the reviewers can have a look on the updated version.

Sincerely,
Authors

---

### Decision · Action_Editor_PNV6 · 2024-10-14

**Recommendation:** Accept with minor revision

**Comment:**

The AE recommends acceptance with a minor revision. Authors have already posted a revision with red text indicating the changes from responding to reviewer concerns. Please integrate these changes for the camera ready and make the data requirements a more visible assumption of the proposed method -- e.g., by extending the 2nd paragraph on page 2 to be clearer about the "two domains" described there.

**Audience:**

All reviewers indicate the is likely an audience for this work at TMLR and the AE agrees. Although adversarial domain adaptation techniques like adapted here have been around for a while, the application to offline RL is interesting and the extended V-D4RL dataset can be a community asset.

**Claims And Evidence:**

After the author response period, all reviewers have indicated that that the manuscript's claims have been suitably supported by empirical evidence.

As reviewer R56R notes, however, the proposed method has some access to data unavailable to other methods in the main experiments. Authors have already adjusted some of this text in the experimental setting to make this clearer, but it is worth making this assumption explicitly in the introduction. Empirically, the additional experiments presented in Appendix D are helpful in clarifying the effect of this assumption on other methods.

---

> ### Author Response · Authors · 2024-10-26
> **Dear Action Editor**
>
> Dear Action Editor,
>
> We have revised the manuscript that integrated reviewers requests, and also more explicitly specified the assumption of data requirements. Furthermore, we have release our codebase in full.